# PLANNING IN NATURAL LANGUAGE IMPROVES LLM SEARCH FOR CODE GENERATION

**Evan Wang**[† 2]    **Federico Cassano**[† 3,4]    **Catherine Wu**[† 5]    **Yunfeng Bai**[1]    **William Song**[1]
**Vaskar Nath**[1]    **Ziwen Han**[1]    **Sean Hendryx**[1]    **Summer Yue**[1]    **Hugh Zhang**[1]
[1]Scale AI    [2]California Institute of Technology    [3]Anysphere
[4]Northeastern University    [5]Anthropic, [†] *work done while at Scale AI*
Correspondence to `ezwang@caltech.edu` and `hugh.zhang@scale.com`

## ABSTRACT

While scaling training compute has led to remarkable improvements in large language models (LLMs), scaling inference compute only recently began to yield analogous gains. We hypothesize that a core missing component is a lack of diverse LLM outputs, leading to inefficient search due to models repeatedly sampling highly similar, yet incorrect generations. We empirically demonstrate that this lack of diversity can be mitigated by searching over candidate plans for solving a problem in natural language. Based on this insight, we propose PLANSEARCH, a novel search algorithm which shows strong results across HumanEval+, MBPP+, and LiveCodeBench (a contamination-free benchmark for competitive coding). PLANSEARCH generates a diverse set of observations about the problem and uses these observations to construct plans for solving the problem. By searching over plans in natural language rather than directly over code solutions, PLANSEARCH explores a significantly more diverse range of potential solutions compared to baseline search methods. Using PLANSEARCH on top of Claude 3.5 Sonnet achieves a pass@200 of 77.0% on LiveCodeBench, outperforming both the best pass-rate achieved without any search (pass@1 = 41.4%) and using standard repeated sampling on top of existing non-search models (pass@200 = 60.6%). Finally, we show that, across all models, search algorithms, and benchmarks analyzed, we can accurately predict performance gains from search as a function of the diversity over generated ideas.

## 1 INTRODUCTION

The bitter lesson (Sutton, 2019) famously posits that two forms of scaling trump everything else: learning and search. While recent advances in large language models (LLMs) have shown that learning is extremely effective, search has not yet proven its value for LLMs, despite its success with classical machine learning techniques (Campbell et al., 2002; Silver et al., 2016; 2017; Brown & Sandholm, 2018; 2019; Bakhtin et al., 2022; FAIR et al., 2022).

Here, we refer to search as any method of spending compute at inference time to improve overall performance (McLaughlin, 2024). In this work, we focus our efforts on improving LLM search for code generation, one of the most important current applications of LLMs. We hypothesize the major bottleneck preventing widespread use of search at inference time for code is a lack of high-level diversity in model outputs. This lack of diversity may arise since common post-training objectives typically do not emphasize generating a diverse set of correct answers, implicitly favoring one correct answer (Rafailov et al., 2024; Ouyang et al., 2022). We empirically demonstrate that this is the case for many open-source language models which have undergone significant post-training. Specifically, we show that in many cases, despite instruction tuned models outperforming base models by large margins on a single sample regime (pass@1), this trend disappears—sometimes even reversing—on a multi-sample regime (pass@k). We refer to Figure 30 as an example of this phenomenon.

Furthermore, the lack of diversity is particularly harmful for search algorithms. In the most egregious of cases with little to no diversity, such as greedy decoding, repeated sampling from the model returns highly similar programs, resulting in minimal gain from additional inference-time compute. This

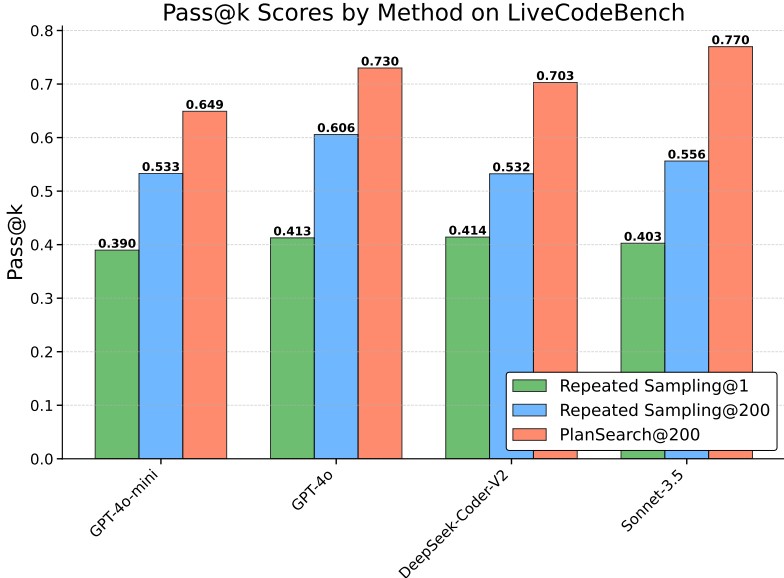

Figure 1: Comparison of REPEATED SAMPLING, both pass@1 and pass@k, and our novel method PLANSEARCH. On every model, our method outperforms baselines by a wide margin, with the best model-method combination of Claude 3.5 Sonnet / PLANSEARCH achieving performance nearly double that of the best model without search.

diversity problem is also not reflected in many public leaderboards (e.g. LMSYS Chatbot Arena (Chiang et al., 2024), LiveCodeBench (Jain et al., 2024), OpenLLMLeaderboard (Aidar Myrzakhan, 2024)), which often report only the pass rate from a single sample of the model, ignoring an entire dimension along which to compare models. While the performance of one sample is the primary metric of relevance for applications such as chatbots, as users typically are sensitive to latency, this single scalar is insufficient to fully capture the quality of a model when it is allowed to use more inference-time compute.

In this paper, we explore several directions for improving the diversity of LLMs at inference time. We hypothesize that the right axis of diversity to search over is the natural language conceptual/idea space, and we validate our hypothesis across several experiments. First, we show that models can produce the correct final program when fed correct solution sketches, where these sketches have been "backtranslated" from passing solution code into sketches in idea space (Section 3.2). Second, we show that when models are asked to generate their own ideas before implementing them on LiveCodeBench (IDEASEARCH), their accuracy conditioned on a particular sketch trends towards either 0% or 100%, suggesting that most of the variance in passing a particular problem is captured by whether the sketch is correct rather than any other factor. These two experiments suggest a natural method to improving LLM search for code generation: by searching for the correct idea to implement.

Guided by this principle of *maximizing exploration of ideas*, we propose PLANSEARCH. In contrast to many existing search methods that search over individual tokens (Zhang et al., 2024; 2023), lines of code (Kulal et al., 2019), or even entire programs (Li et al., 2022), PLANSEARCH searches over possible *plans* for solving the problem at hand, where a plan is defined as a collection of high level observations and sketches helpful to solve a particular problem (Figure 2). To generate novel plans, PLANSEARCH generates a number of observations about the problem, before combining these observations into a candidate plan for solving the problem. This is done for every possible subset of the generated observations to maximally encourage exploration in idea space, before the codes are eventually all translated into a final code solution (Section 4.3). We find that searching over plans outperforms both standard repeated sampling and directly searching over ideas (IDEASEARCH, introduced in Section 4.2) in terms of effectively using compute at inference time.

Applying PLANSEARCH on top of Claude 3.5 Sonnet achieves a pass@200 of 77.0% on Live-CodeBench, outperforming both the best score achieved without search (pass@1 = 41.4%) and

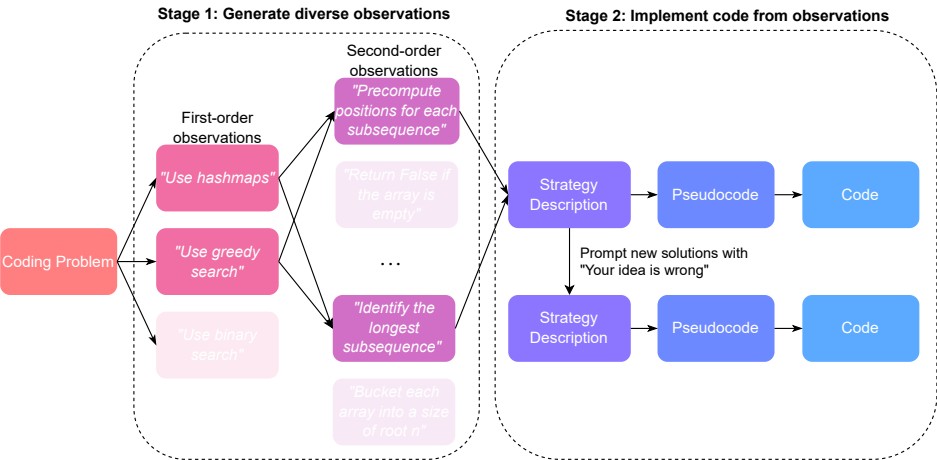

Figure 2: An example trajectory of PLANSEARCH, which searches over plans in natural language as a method of increasing diversity in the search process. PLANSEARCH first generates observations, then combinatorially samples subsets of these observations to generate the next step in the search process. To generate the next layer of observations, the combinations derived from the first observations are used as a stepping stone to generate the next observations, and the process repeats. After generating both the first and second order observations, PLANSEARCH then generates a natural language description of a strategy to solve the problem. For additional diversity, the model is prompted to regenerate its strategy as an additional sample before generating code. See Section 4.3 for additional discussion.

the standard best-of-n sampling score on non-search methods (pass@200 = 60.6%). Furthermore, consistent with recent findings on the effectiveness of search on top of small models (Chen et al., 2024; Brown et al., 2024; Bansal et al., 2024; Wu et al., 2024), running PLANSEARCH on top of a small model (GPT-4o-mini) outperforms larger models not augmented with search after merely 4 attempts. Evaluations of PLANSEARCH across two other coding benchmarks, HumanEval+ and MBPP+ (Liu et al., 2023), suggest similar improvements.

Finally, we measure the diversity of output code over the idea space of all search methods via an LLM-as-a-judge procedure (Section 6.1) and show that the resulting diversity score is highly correlated with the performance gains generated by that search method. This provides further support for our hypothesis that the effective exploration of plans in idea space is key to LLM search for code generation (Figure 5).

## 2 RELATED WORK

We reiterate that search as defined in the context of our paper refers to any method which expends inference-time compute to improve performance. We further specify planning as any form of high level observation or abstract thought that assists a model in generating a final solution. Our work builds off a long history of work in scaling search and planning. For information on relevant work in classical AI, general search, and filtering, see Appendix Q.

Regarding searching over plans in natural language, several approaches have proposed generalizing chain-of-thought (Wei et al., 2022) reasoning into a search-like process, such as Tree of Thoughts (Yao et al., 2023a) and Reasoning via Planning (Hao et al., 2023b). However, prior methods have largely demonstrated effectiveness on somewhat contrived problems designed to highlight the power of search, such as the game of 24, or classic planning benchmarks such as Blocksworld (McDermott, 2000), where both benchmarks are easier to solve by explicitly considering many options, and where the 'steps' over which to search over are fairly obvious. By contrast, most real-world planning is used to assist in domains that are complex enough to benefit from, but not require, the additional exploration of plans. We demonstrate that PLANSEARCH, which plans in natural language, outperforms baseline

search methods in one such domain: code generation. Moreover, our analysis reveals the underlying reason that such search is effective: it increases the diversity of the generated ideas, allowing more efficient search relative to other methods which repeatedly submit highly similar, incorrect solutions. This is consistent with prior work suggesting the importance of diversity in natural language generation (Hashimoto et al., 2019; Zhang et al., 2021). Other directions for search include decomposing programs down into smaller parts before solving each one individually (Zelikman et al., 2023; Zhou et al., 2023; Gao et al., 2023).

PLANSEARCH is also distinct from other methods which explicitly train a model to search or on reasoning traces sampled from the model (Zelikman et al., 2022; Zhang & Parkes, 2023; Zelikman et al., 2024; OpenAI, 2024) in that PLANSEARCH induces diversity at inference-time and converts an LLM API not designed for search into one that is capable of showing strong gains from search. Separately, there is a large family of work in the agent space, in which outputs from terminal commands or other tools are fed back into the model before the agent is queried for the next step (Shinn et al., 2023; Yao et al., 2023b; Schick et al., 2023; Chen et al., 2022b).

## 3 MOTIVATION

Coding is a powerful area in which search should excel. While search in other domains requires both generating many solutions *and* selecting the correct solution amongst all the resulting generations, coding often only requires the former, as any valid piece of code can be tested via code execution against given test cases. This allows code search algorithms to sidestep many of the issues that plague search algorithms for more open-ended domains (e.g. generating poetry) due to difficulty in selecting correct solutions out of all the generated solutions.

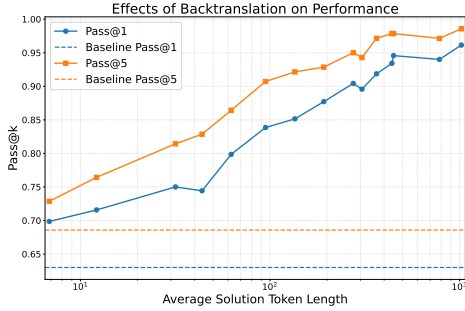
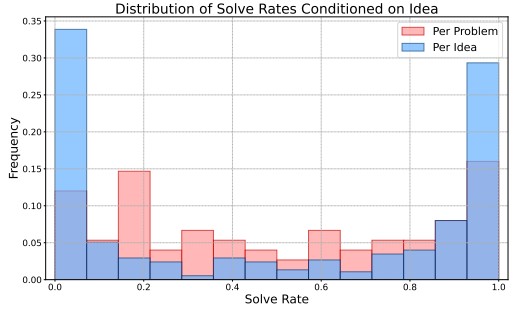

(a) Performance of GPT-4o-mini on LiveCodeBench when provided with backtranslated solutions of varying lengths. The baselines plot performance without backtranslated solutions. Providing the model with a compressed solution in natural language, even as short as 10 tokens, significantly increases performance.

(b) We plot the distribution of solve rates conditioned on being given a solution sketch and without. When conditioning on a given sketch, we notice that downstream solve rates polarize towards either 0% or 100%. Most of the variance in performance is predicted by whether a given idea is correct or not.

Figure 3: Backtranslation shows the promise of providing good sketches, and conditioning on idea shows the presence of a solution sketch polarizes performance.

### 3.1 DEFINING THE SEARCH SPACE

Perhaps the most important question for eliciting strong search capacities is determining which space to search over, as finding the proper layer of abstraction is critical to progress in the field. Prior approaches have varied, with many people searching over individual tokens (Zhang et al., 2024; 2023), lines of code (Kulal et al., 2019), or even entire programs (Li et al., 2022). We hypothesize that the key factor is obtaining *the correct solution sketch*, which we define as a description of the correct program in natural language space. Intuitively, conducting the reasoning process in natural language space allows us to effectively harness the training process of LLMs, which have observed many human reasoning traces in both pre- and post-training. Prior work (Wei et al., 2022) has observed strong positive effects from being allowed to conduct such reasoning in natural language, making it a natural place to search over. We describe two experiments providing evidence for this hypothesis by testing on the LiveCodeBench benchmark using GPT-4o-mini as our model.

## 3.2 BACKTRANSLATION

To investigate the hypothesis whether the idea space, instantiated as solution sketches, is the right area of exploration, a natural question is whether LLMs can correctly implement a correct code solution given a correct sketch. Inspired by approaches to backtranslation in machine learning (Sennrich et al., 2016; Pham et al., 2021; Edunov et al., 2018), we experiment with "backtranslating" passing code solutions back into idea space. First, we generate code solutions using GPT-4o to generate 1000 attempts to solve the problem and filter out problems without any passing solutions. As we also do not have a dataset of correct solution sketches associated with each solution, we generate a candidate correct idea via backtranslation. We do this by feeding an LLM both the problem and code solution and asking the LLM to convert said solution into a natural language description of the solution. Additionally, we vary the detail of the backtranslated idea via instructions to the LLM in the prompt (e.g. 'in $w$ words'). A full description of the prompts can be found in Appendix M.1, alongside several example backtranslated solutions of various lengths.

We observe that prompting a model with a backtranslated idea significantly improves accuracy, increasing with the length of the translated idea (Figure 3a), which suggests that having a correct sketch is sufficient to produce the correct final solution with relatively high accuracy, even only after 10 tokens of backtranslated solution. This suggests that the correct direction of search is to explore through idea space to maximize the chance of arriving at a correct idea.

## 3.3 CONDITIONING ON IDEA QUALITY

In a follow-up experiment, we prompt an LLM to generate its own sketches to solve LiveCodeBench problems instead of providing it with golden ones via backtranslation. First, we generate 5 ideas per problem using IDEASEARCH, defined in Section 4.2. For each idea, we then sample 25 candidate solutions and measure their pass rate. For this experiment, we filter out any problem that GPT-4o-mini solves with either a $100\%$ or a $0\%$ solve rate, since such problems are either too easy or too hard for the model and would not be informative for this experiment. We end with 75 problems and 375 sketches.

To test our hypothesis that generating a correct sketch is a critical factor for solving problems, we compare the distribution of solve rates for generating correct code solutions *conditioned* on a given sketch to the distribution over solve rates given a sketch drawn at random, i.e., just the distribution over solve rates. While verifying whether a sketch is correct or incorrect is difficult without access to external labels, a key insight is that if generating the correct idea is a critical factor in solving the problem, then conditioning on a particular sketch should polarize the distribution of solve rates towards $\{0, 1\}$. If the model is given a correct sketch, it should consistently generate correct solutions, while if given a bad sketch, it should consistently generate incorrect solutions.

Our results confirm this to be the case. Figure 3b shows the distribution of solve rates across problems, both unconditionally (in red) and conditioned on each sketch (in blue). We notice that when grouping by sketches, the solve rates indeed become polarized towards $\{0, 1\}$. This result has important implications for improving code generation, suggesting that a large portion of variance in performance can be explained by whether the model is able to generate a correct idea or not. Therefore, a natural path for improvement is to focus on the sketch generation step and search for correct sketches and observations in idea space before generating solution code.

## 4 METHODS

We provide a description of the various methods of search we explore in our work. If additional background on competitive programming and related notation is desired, we provide more (optional) information in Appendix P.

## 4.1 REPEATED SAMPLING

We consider the basic prompting approach as a baseline, in which we use few-shot prompting by providing the LLM with a number of problem-solution pairs before asking it to solve the desired question (Brown et al., 2020). A full example of the prompt is given in Appendix M.2. In code

generation, the most common variant of search utilized is repeated sampling, where models are repeatedly sampled from until they generate an output that passes the test or the maximum number of samples is reached. Refer to the Related Work for more information (Section Q.2).

## 4.2 IDEASEARCH

A natural extension of the REPEATED SAMPLING approach discussed in Section 4.1 is to avoid prompting the LLM for the solution code immediately. This can be viewed as an application of the commonly used "chain-of-thought" prompting to programming problems (Wei et al., 2022), although we find that IdeaSearch shows non-negligible performance boosts over standard "chain-of-thought" prompting (see Appendix E).

In IDEASEARCH, the LLM is given the problem $P$ and is asked to output a natural language solution $S$ of the problem. Then, a separate instance of the LLM is given $P$ and $S$, and tasked to follow the proposed solution $S$ to solve the problem $P$. The purpose of IDEASEARCH is to isolate the effectiveness of having the correct "idea/sketch" for solving the problem. Empirically, we find that explicitly forcing the search algorithm to articulate an idea for solving the problem increases diversity. See Appendix M.3 for detailed prompts.

## 4.3 PLANSEARCH

While both REPEATED SAMPLING and IDEASEARCH are successful and lead to improvement in the results on benchmark results, we observe that in many of the cases, prompting multiple times (pass@k) (even at high temperatures) will only lead to small, narrow changes in the output code that change minor aspects but fail to improve upon pitfalls in idea.

Ablations for many of the choices in the subsequent description of PLANSEARCH can be found in Appendix H.

### 4.3.1 PROMPTING FOR OBSERVATIONS

Starting from the problem statement $P$, we prompt an LLM for "observations"/hints to the problem.

We denote these observations as $O_i^1$, where, $i \in \{1, \ldots, n_1\}$ due to the fact that they are first-order observations. Typically, $n_1$ is on the order of 3 to 6. The exact number depends on the LLM output. To use these observations to inspire future idea generation, we create all subsets with size at most $S = 2$ of $s^1 = \{O_1^1, \ldots, O_{n_1}^1\}$. Each of these subsets is a combination of observations, and for clarity we denote each subset as $C_i^1, i \in \{1, \ldots, l_1\}$, where $l_1 = 1 + n_1 + \binom{n_1}{2}$.

### 4.3.2 DERIVING NEW OBSERVATIONS

The set of all observations can be thus defined as a directed tree with depth 1, where the root node is $P$, and an edge exists for each $C_i^1$ pointing from $P$ to $C_i^1$. We then repeat this procedure from Section 4.3.1 on each leaf node $C_i^1$ to generate a set of second order observations, $s_i^2 = \{O_{i,1}^2, \ldots, O_{i,n_{i,2}}^2\}$. To obtain second order observations, we prompt the model with both the original problem $P$ and all observations contained in $C_i^1$, framed as primitive observations that are necessary in order to solve $P$. The LLM is then prompted to use/merge the observations found in $C_i^1$ in order to derive new ones.

The same procedure as Section 4.3.1 is used to create all subsets $C_{i,j}^2$, for all $i \in \{1, \ldots, l_1\}$. This process may be arbitrarily repeated, but we truncate the tree at depth $L = 2$ for computational constraints.

Note that there is no assumption any of the observations generated are correct. In fact, it is critical to note that many of them may be incorrect. The observations merely serve to elicit the model to search over a more diverse set of ideas.

### 4.3.3 OBSERVATIONS TO CODE

After the observations have been made, they must be implemented as ideas before being translated into code. For each leaf node, we prompt the model with all observations, along with the original

problem $P$, in order to generate a natural language solution to the problem $P$. To add more diversity, for each generated idea, we generate an additional idea by supposing the idea is wrong, and asking an LLM to give criticisms/feedback, thus increasing our proposed ideas by a factor of 2.

These natural language solutions are then translated into pseudocode, which are subsequently translated into actual Python code. We take a more granular approach to reduce the translation error (which may cause the model to revert to its original mode, disregarding the reasoned-through observations). We provide all prompts for all sections in Appendix M.4.

## 5 EXPERIMENTAL RESULTS

| Model | Eval | Pass@1 | Pass@200 | IS@200 (ours) | PS@200 (ours) |
|---|---|---|---|---|---|
| GPT-4o-mini | LCB | 39.0 | 53.3 | 59.4 | 64.9 |
| GPT-4o | LCB | 41.3 | 60.6 | 70.4 | 73.0 |
| DeepSeek-Coder-V2 | LCB | 41.4 | 53.2 | 65.9 | 70.3 |
| Claude-Sonnet-3.5 | LCB | 40.3 | 55.6 | 70.2 | 77.0 |
| GPT-4o-mini | HE+ | 83.7 | 95.0 | 97.5 | 98.2 |
| GPT-4o | HE+ | 86.4 | 98.2 | 97.6 | 99.5 |
| DeepSeek-Coder-V2 | HE+ | 82.8 | 91.4 | 97.2 | 99.3 |
| Claude-Sonnet-3.5 | HE+ | 81.6 | 88.9 | 95.6 | 98.5 |
| GPT-4o-mini | M+ | 73.5 | 83.8 | 87.3 | 91.0 |
| GPT-4o | M+ | 77.2 | 87.4 | 89.3 | 92.2 |
| DeepSeek-Coder-V2 | M+ | 76.3 | 81.9 | 89.1 | 92.6 |
| Claude-Sonnet-3.5 | M+ | 77.1 | 83.0 | 87.8 | 93.7 |
| o1-mini (search model) | LCB | 69.5 | 90.8 | 91.2 | 91.3 |

Table 1: LCB, HE+, M+ short for LiveCodeBench, HumanEval+, and MBPP+, respectively. IS short for IDEASEARCH and PS short for PLANSEARCH. We find that PLANSEARCH and IDEASEARCH improve upon search baselines across all models, with PLANSEARCH achieving the best results across all models and benchmarks considered. Notably, using PLANSEARCH on top of Claude 3.5 Sonnet (Anthropic, 2024) has a pass@200 of 77.0 on LiveCodeBench, which is nearly double the performance of the top model without using search (41.4). PLANSEARCH also outperforms basic pass@200 on o1-mini for LiveCodeBench, though since o1-mini already uses inference-time compute, the gap is much smaller than compared to non-search models. The full pass@k curves are included in Appendix A.

### 5.1 DATASETS

We evaluate our search methods on three benchmarks: MBPP+, HumanEval+ (Liu et al., 2023), and LiveCodeBench (Jain et al., 2024). MBPP (Austin et al., 2021) and HumanEval (Chen et al., 2021) are some of the most widely used code benchmarks in the field. However, since both benchmarks provide only a few test cases, Liu et al. (2023) updates both benchmarks with additional test cases that increase the benchmarks' robustness to reward hacking. LiveCodeBench is a benchmark for coding that consists of competitive programming problems which typically require advanced reasoning capabilities. Given the reality that coding data is often highly upsampled during pre-training (OpenAI et al., 2024; Dubey et al., 2024), LiveCodeBench differentiates itself from other benchmarks by taking care to segregate problems by date to avoid data contamination concerns. For this paper, we use only the subset of problems between May 2024 and September 2024 to avoid possibilities of contamination. We choose May 2024 as the cutoff date to ensure that our results with our best performing model (Claude 3.5 Sonnet) are not due to contamination, because Claude 3.5 Sonnet has a knowledge cutoff of April 2024. To ensure fair comparison, we use the same cutoff for all models evaluated, even though the precise cutoff dates for other models may vary slightly from May 2024.

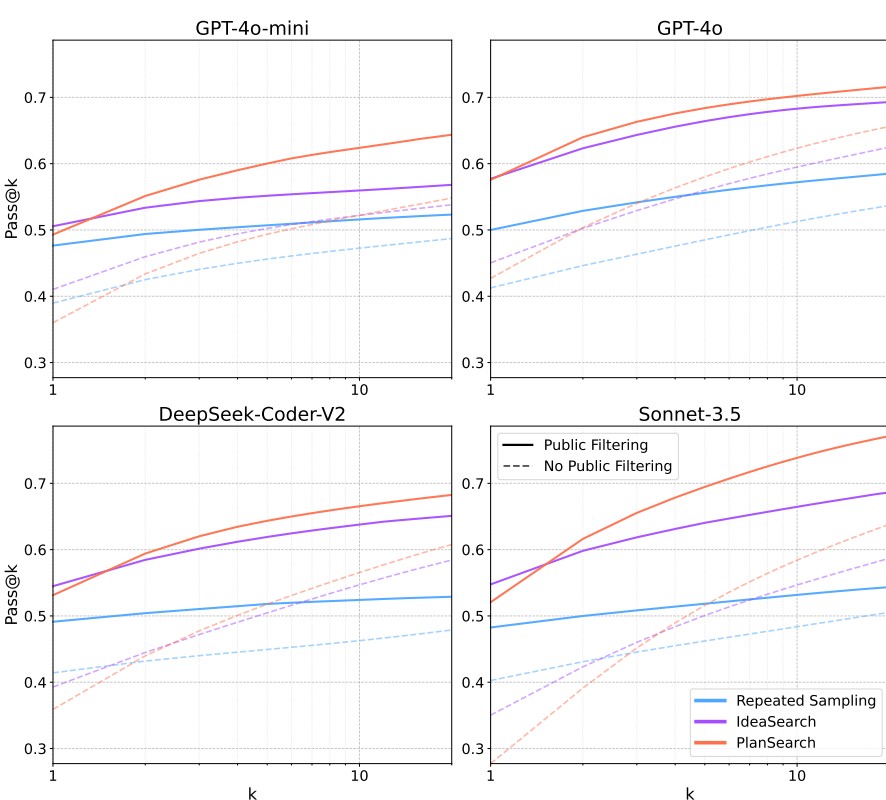

Figure 4: Performance of all models and methods on LiveCodeBench with public test filtering. The purpose of filtering is to shift pass@k curves leftward (i.e., bringing performance at high $k$ to low $k$), so we plot curves in detail over $k \in \{1, \ldots, 20\}$. Even at 10 completions, PLANSEARCH outperforms *filtered* REPEATED SAMPLING by a flat 30 to 40%. Again, full pass@k plots are included in their entirety in Appendix A.

## 5.2 EXPERIMENT DETAILS

For all search algorithms, we require that all output code be in the correct format specified, and we mark a solution as incorrect if it does not follow the intended formatting. The extracted code is then run through all tests of the program and marked as correct if and only if it passes all tests.

All models are run with temperature $0.9$ and top-$p$ of $0.95$. (o1-mini was run with temperature $1.0$ and top-$p$ of $1.0$ because of API constraints.) Temperature was determined through a coarse hyper-parameter sweep on REPEATED SAMPLING and IDEASEARCH from $T \in \{0.0, 0.1, 0.2, \ldots, 1.2\}$, which we describe in Appendix F.

Both REPEATED SAMPLING and IDEASEARCH generate exactly $n$ codes, whereas PLANSEARCH generates a variable number of codes, usually ranging on the order of 300 to 400. To compute pass@k, we use the unbiased estimator in Equation 4 (Chen et al., 2021)[1].

If $k > n$, we assume the remaining generations did not pass. To compute pass@k for filtering, we limit the pool of codes to those that are filtered, meaning that both $n$ and $c$ may shrink in size. This can be thought of as a conditional probability, where the condition is that the code passes public tests. For more information on public test filtering, see Appendix R.

---

[1]Note that the estimator in Equation 4 theoretically requires that the number of successes follows a binomial distribution. REPEATED SAMPLING and IDEASEARCH obey this, but PLANSEARCH generations may not be independent. See Appendix O for more discussion.

## 5.3 RESULTS

Our summarized results for REPEATED SAMPLING, IDEASEARCH, and PLANSEARCH can be found in Table 1, Figure 1, and Figure 4. We find that PlanSearch improves over existing methods for all models and benchmarks considered.

Additionally, we plot our full pass@k curves for all methods, models, and datasets in Appendix A. Due to prohibitively high costs, we only evaluate o1-mini on LiveCodeBench, as HumanEval+ and MBPP+ show strong saturation effects even with weaker models like GPT4o-mini. The pass@k curves of o1-mini and associated discussion can be found in Appendix K. For sake of easy comparison, we also plot all relative gains compared to REPEATED SAMPLING@1 averaged over all models in Appendix C. For a compute-normalized comparison between REPEATED SAMPLING and PLANSEARCH, see Figure 17. Additionally, we ablate over the design choices made for PLANSEARCH in Appendix H.

## 6 ANALYSIS

Our results suggest that both PLANSEARCH and IDEASEARCH outperform basic sampling by a wide margin (Figures 11, 12, 13), with PLANSEARCH achieving the best score across all methods and models considered. Since o1-mini is unique from all other models tested, we show and discuss its unique results in Appendix K. We show the detailed pass@k results for each dataset in Figures 6, 7 and 8. We also compare with Chain-of-Thought (Wei et al., 2022) in Appendix E. Interestingly, we find that IDEASEARCH performs somewhat better, which we speculate comes from differences in splitting solution sketch into *two* model responses, instead of doing both chain-of-thought and code solution in one model response.

Investigating the differences in specific models, we notice that trends exhibited by the pass@k curves are not uniform across all models; in fact, each curve seems unique. We hypothesize that these differences are in part due to changes in idea diversity, as investigated in Figures 5, 25, 26. Figure 5 includes o1-mini diversities, which also follow the observed trend. From the figures, we can see that our approximate diversity score accounts for much of the variance we see in the relative improvement that arrives from scaling-up inference-time compute. This correlation holds across all methods and models on the same dataset, thus suggesting that diversity score can be used as a proxy to predict for relative pass@k improvement. For further discussion on the specifics of the diversity score, see Section 6.1.

One interesting point of observation is that PLANSEARCH often hurts pass@1 for several models, including most notably Sonnet 3.5 on LiveCodeBench, our best performing combination. Intuitively, this is because increasing the diversity across ideas likely dilutes the probability that any *particular* idea is generated, while simultaneously increasing the chance of having *at least one* correct idea within said pool. Therefore, pass@1 may be slightly lower than usual, yet pass@k will likely surpass "pools" of ideas lacking diversity for this reason. See Figure 41 for a graphical intuition.

Finally, in Table 1 and Figure 1, we present our main results normalized across attempts/completion, where each search method is allowed $k$ attempts to solve each problem. An alternative method of normalizing across methods is to equalize the amount of compute spent on each method. Since PLANSEARCH and IDEASEARCH first plan out an idea before implementing the final solution, they both spend more compute at inference time per solution generated. In Appendix D, we report the equivalent plots normalized across compute. Our findings are highly similar and suggest that PLANSEARCH outperforms all other methods if sufficient compute is expended at inference time.

### 6.1 MEASURING DIVERSITY

We find that idea-space diversity strongly predicts search performance, measured by the relative improvement between pass@1 and pass@200 (Figure 5). While entropy is a common diversity measure (Shannon, 1948), it is inadequate for LLM settings (Hashimoto et al., 2019; Zhang et al., 2021). For instance, a model generating variations of the same program and another producing distinct programs may have equal entropy, yet the latter will perform better in search-augmented tasks.

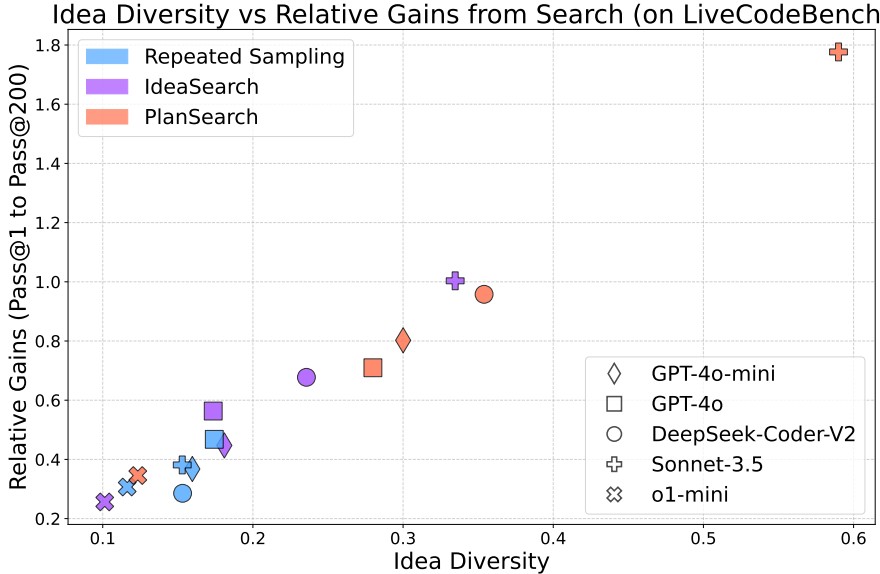

Figure 5: We observe a strong positive correlation between the measured amount of idea diversity (higher score is better) in a search algorithm and the resulting improvements due to search. See Section 6.1 for information regarding the diversity score.

We measure diversity in idea space via pairwise comparisons across all generated programs. Let $\{c_1, \ldots, c_n\}$ be a set of $n$ code generations, each corresponding to a latent idea. Two sketches can be thought to be considered similar if they are within some $\epsilon$ of each other in idea space, noting that transitivity may not hold here.

To compute diversity, we construct the $\binom{n}{2}$ pairs and evaluate their similarity using an LLM, defining $S(c_i, c_j) \in \{0, 1\}$, where $S(c_i, c_j) = 1$ iff $c_i$ and $c_j$ are similar. The diversity score is:

$$D = 1 - \frac{\sum_{i<j} S(c_i, c_j)}{\binom{n}{2}} \tag{1}$$

A model producing all identical ideas has $D = 0$, while one generating all unique ideas has $D = 1$. A score of $D$ is also equivalent to the probability that two randomly selected programs are similar to each other (see Appendix S for more details).

For each method, we report the diversity score across all problems. For large $n$, we sample 40 codes and compare all pairs. GPT-4o-mini is used to evaluate $S$, and full prompt details are in Appendix O.1.

## 7 CONCLUSION

In this work, we find that diversity in idea space is incredibly useful to unlock significant achievements in the effectiveness of inference-time compute—otherwise referred to as search—particularly in code generation tasks. We propose PLANSEARCH, which obtains great performance on all datasets tested, almost doubling baseline performance at pass@200. Additionally, we find strong correlation between our diversity metric and resulting performance gains from evaluating at pass@k instead of pass@1, which underscores the importance of idea diversity in effective search. We believe that these insights can be applied to many other domains and will be crucial to realize the full potential of LLMs to unlock significant performance gains as seen here.

However, while PLANSEARCH substantially improves diversity over idea space at inference-time, fundamentally, improvements in diversity should also come at the post-training stage, like with methods such as o1 (OpenAI, 2024). This likely requires re-imagining the post-training pipeline for LLMs around search.

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

# A FULL PASS@K CURVES FOR ALL MODELS AND ALL BENCHMARKS

See Figures 6, 7, 8. We plot all models and methods on HumanEval+, MBPP+ (Liu et al., 2023), and LiveCodeBench (Jain et al., 2024), respectively.

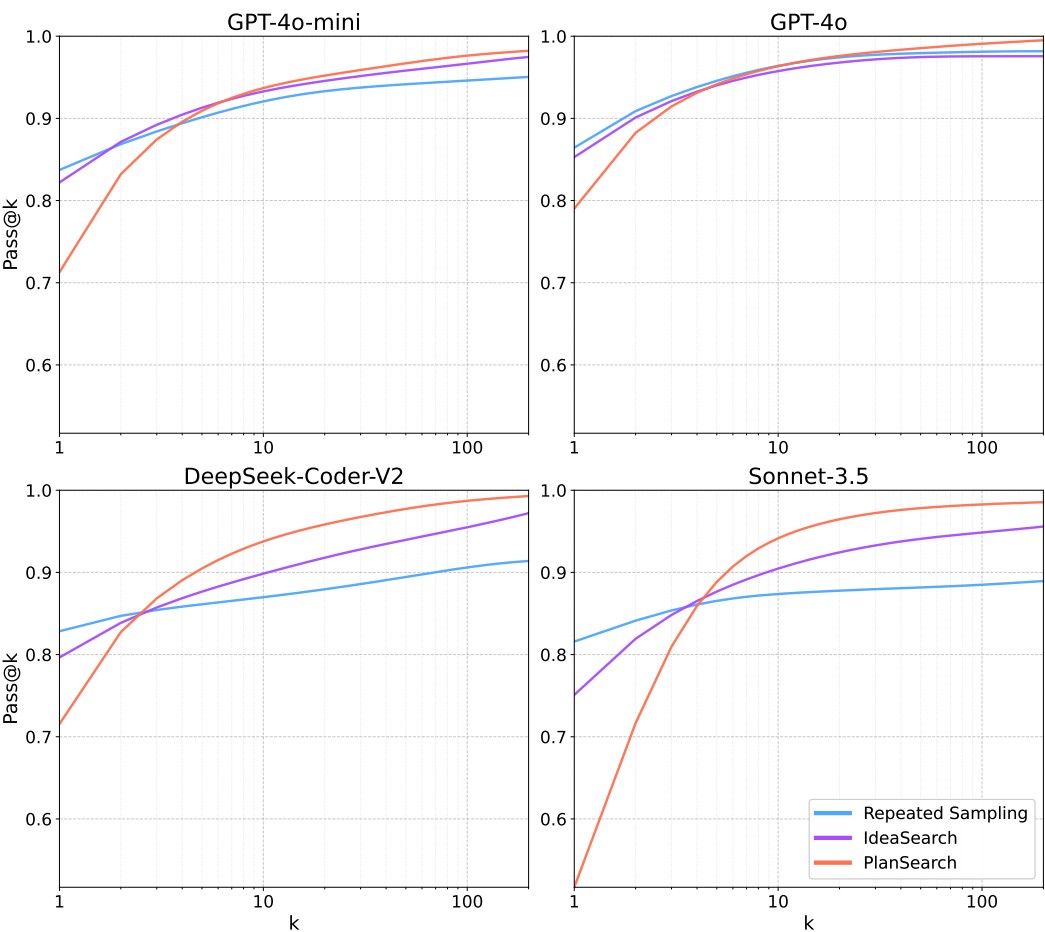

Figure 6: Pass@k performance of all models and methods on HumanEval+, plotted over $k \in \{1, \ldots, 200\}$.

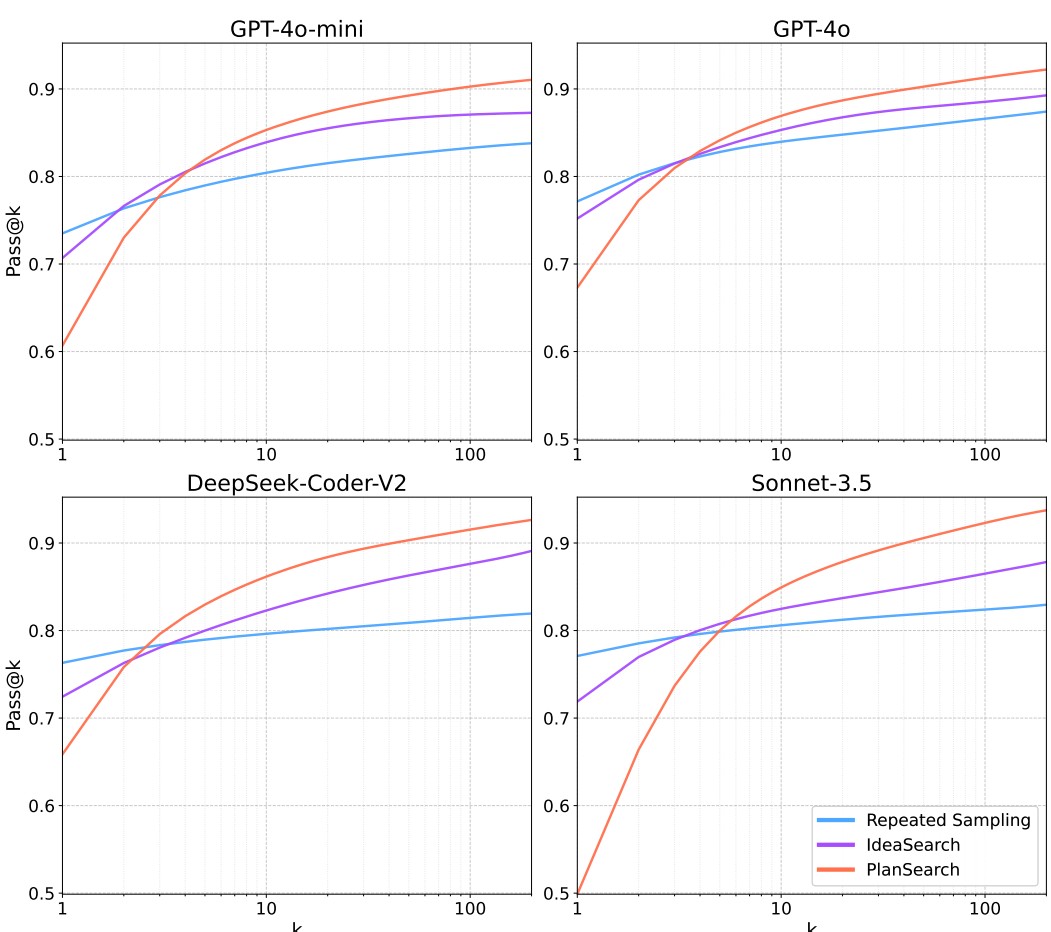

Figure 7: Pass@k performance of all models and methods on MBPP+, plotted over $k \in \{1, \ldots, 200\}$.

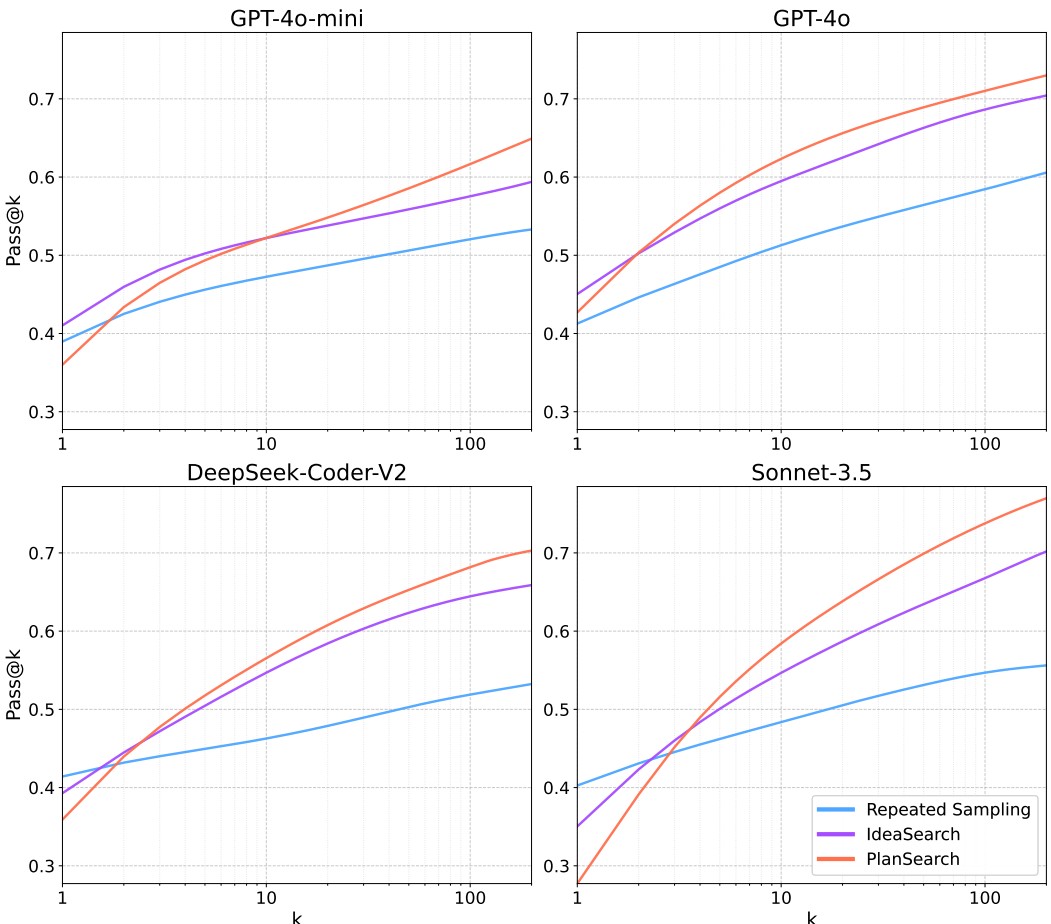

Figure 8: Pass@k performance of all models and methods on LiveCodeBench, plotted over $k \in \{1, \ldots, 200\}$.

# B   FULL PASS@K CURVES WITH PUBLIC FILTERING

See Figures 9, 10, 4. We plot all models and methods with public test filtering on HumanEval+, MBPP+ (Liu et al., 2023), and LiveCodeBench (Jain et al., 2024), respectively.

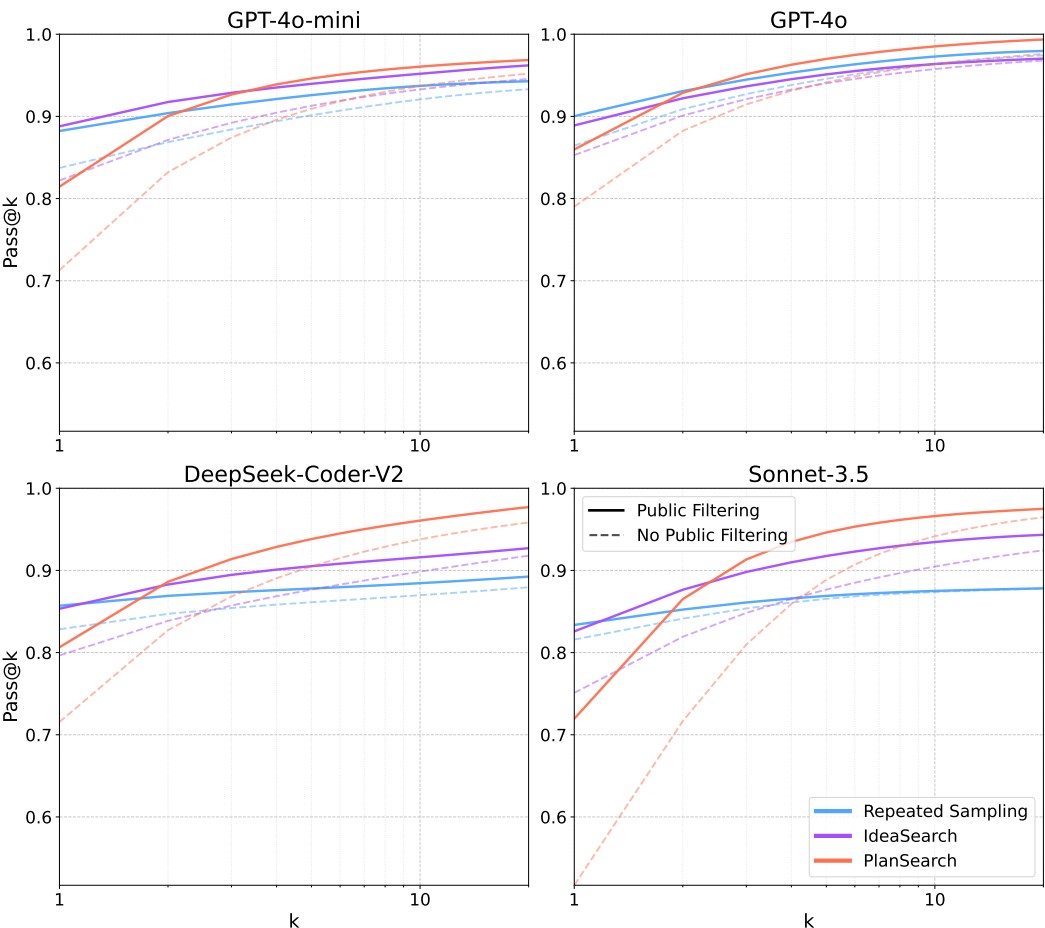

Figure 9: Pass@k performance of all models and methods on HumanEval+, with public test filtering, plotted over $k \in \{1, \ldots, 20\}$. Note that dotted lines are provided for reference of the base method pass@k before filtering.

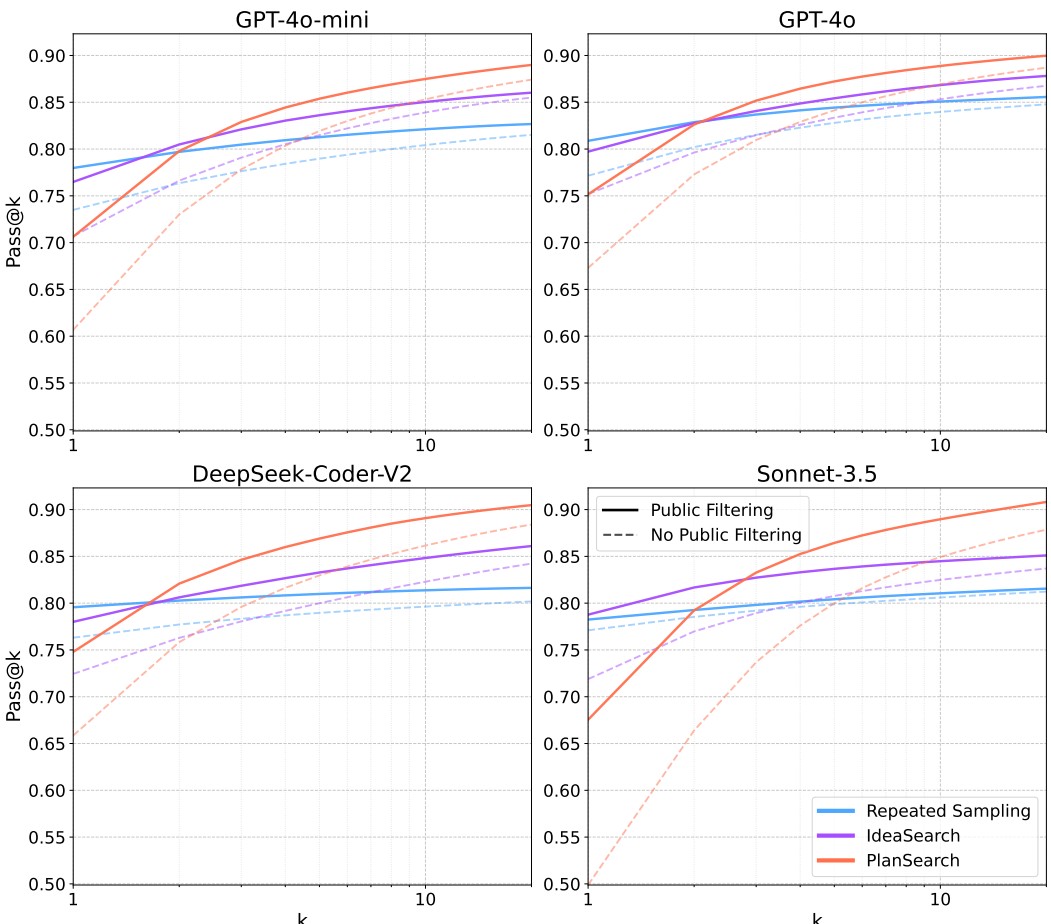

Figure 10: Pass@k performance of all models and methods on MBPP+, with public test filtering, plotted over $k \in \{1, \ldots, 20\}$. Note that dotted lines are provided for reference of the base method pass@k before filtering.

# C  AVERAGE RELATIVE IMPROVEMENTS

See Figures 11, 12, 13. To create these graphs, the relative improvements of each point on all pass@k curves are computed and compared to the respective pass@1 of REPEATED SAMPLING. Then these values are averaged over all models, so that there is one curve per method per dataset. The datasets are HumanEval+, MBPP+ (Liu et al., 2023), and LiveCodeBench (Jain et al., 2024), respectively. For the public test filtered versions, see Figures 14, 15, 16.

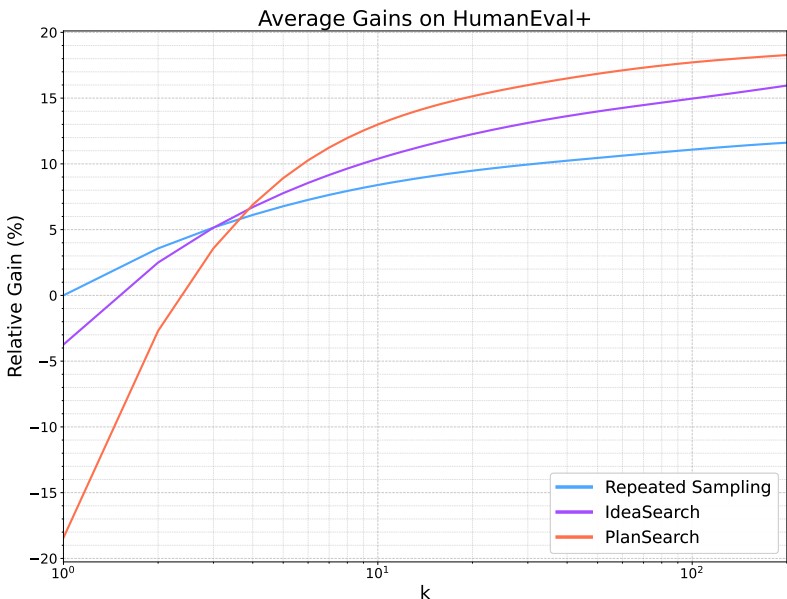

Figure 11: Performance gain over REPEATED SAMPLING@1 averaged over all models on HumanEval+, plotted over $k \in \{1, \ldots, 200\}$.

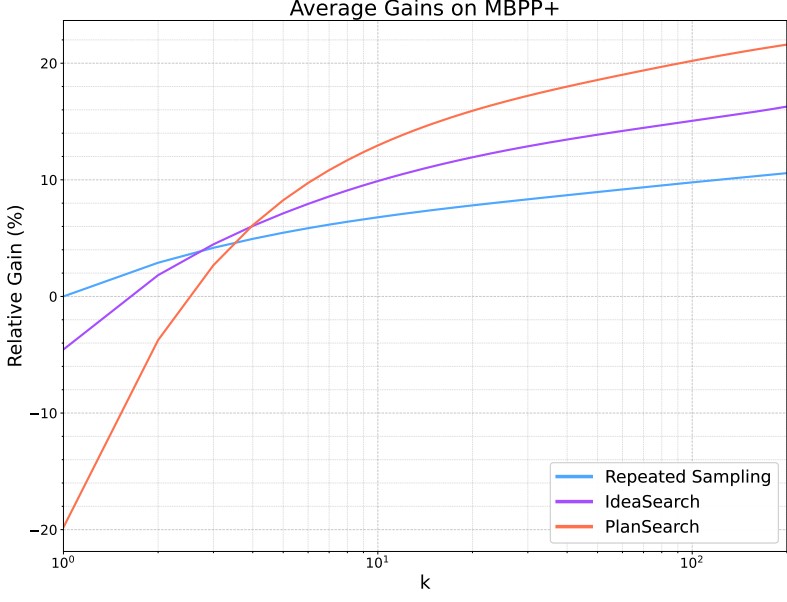

Figure 12: Performance gain over REPEATED SAMPLING@1 averaged over all models on MBPP+, plotted over $k \in \{1, \ldots, 200\}$.

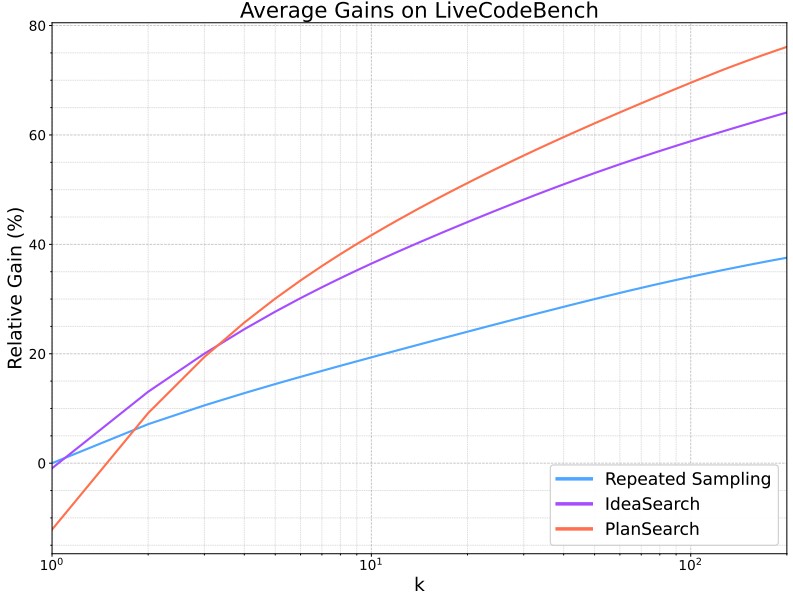

Figure 13: Performance gain over REPEATED SAMPLING@1 averaged over all models on Live-CodeBench, plotted over $k \in \{1, \ldots, 200\}$.

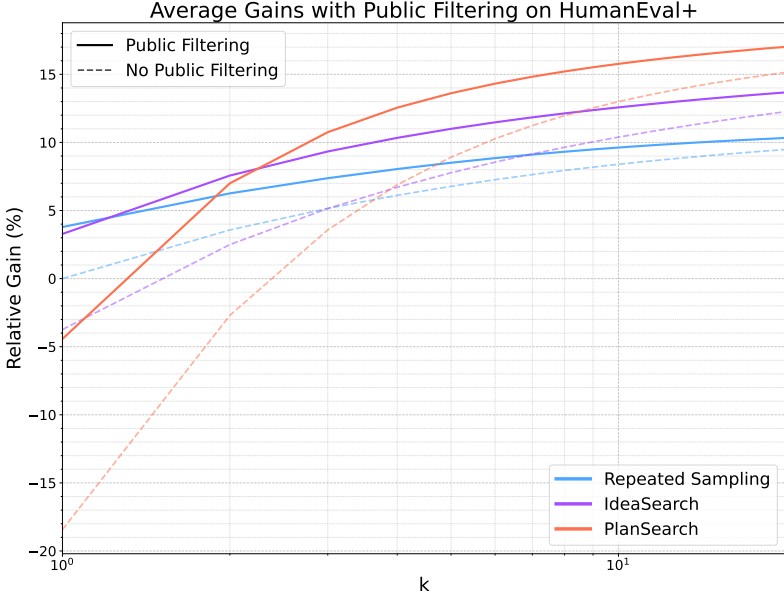

Figure 14: Average performance gain over all models of methods with public test filtering compared to REPEATED SAMPLING@1, plotted over $k \in \{1, \ldots, 20\}$. Note that dotted lines are provided for reference of the base method pass@k (before filtering).

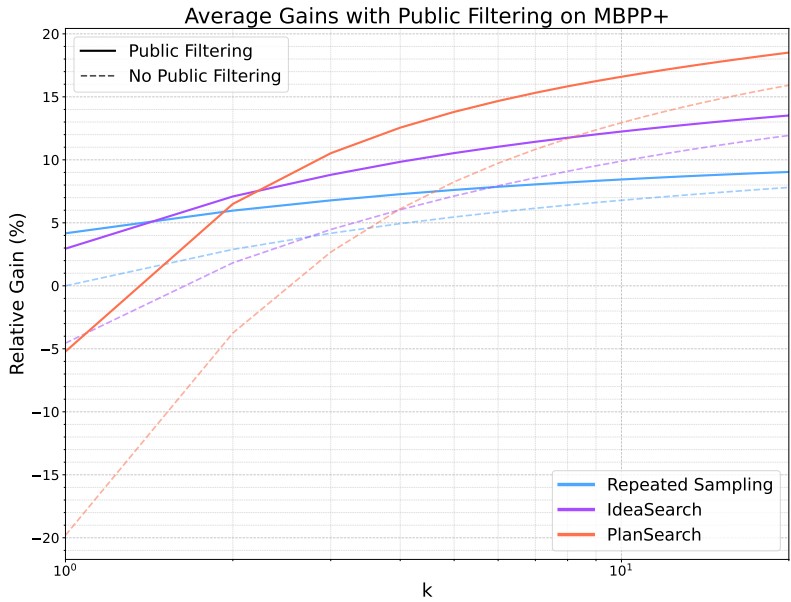

Figure 15: Average performance gain over all models of methods with public test filtering compared to REPEATED SAMPLING@1, plotted over $k \in \{1, \dots, 20\}$. Note that dotted lines are provided for reference of the base method pass@k (before filtering).

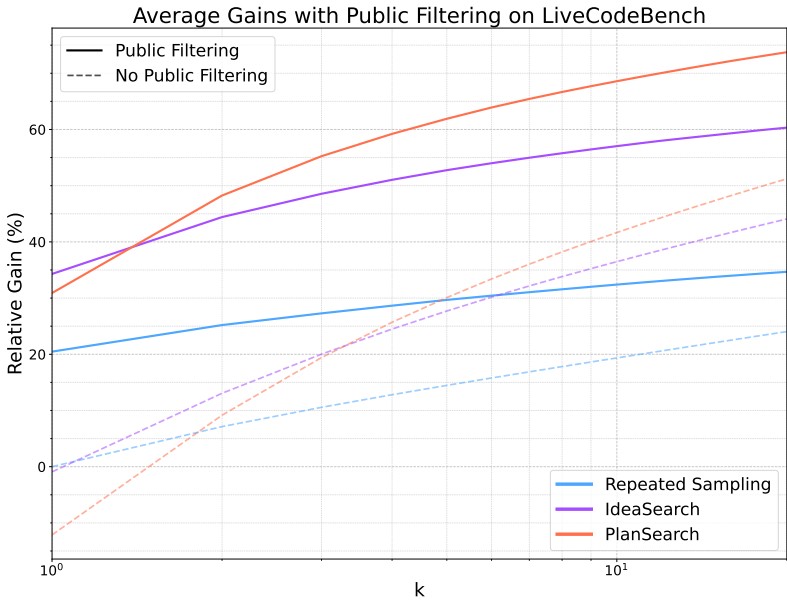

Figure 16: Average performance gain over all models of methods with public test filtering compared to REPEATED SAMPLING@1, plotted over $k \in \{1, \dots, 20\}$. Note that dotted lines are provided for reference of the base method pass@k (before filtering).

# D    COMPUTE NORMALIZED PASS@K GRAPHS

See Figure 17. For each run of a method in Appendix A, we compute the number of generated tokens needed per completion, per problem, independently on each dataset. Then, we average across all datasets to obtain 244 generated tokens per completion per problem for REPEATED SAMPLING, and 1, 428 generated tokens per completion per problem for PLANSEARCH.

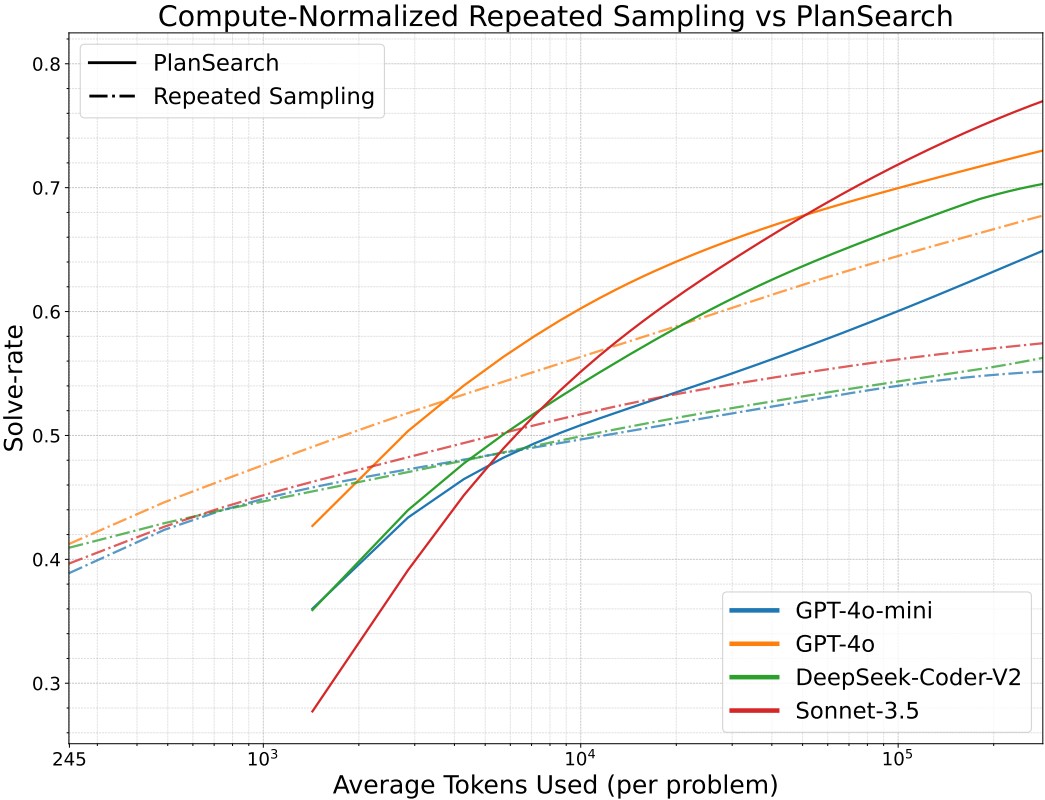

Figure 17: Normalized pass@k by average tokens used per problem. REPEATED SAMPLING uses roughly 244 tokens per completion per problem, and PLANSEARCH uses roughly 1428 tokens per completion per problem. When we normalize compute across methods, we find that PLANSEARCH begins to be more effective than repeated sampling if the user is willing to sample at least 10,000 tokens per problem.

# E COMPARISON WITH CHAIN-OF-THOUGHT

See Figures 18, 19, 20, which are run on LiveCodeBench (Jain et al., 2024), MBPP+, and HumanEval+ (Liu et al., 2023), respectively. These are the same plots as Appendix A, with CoT (Wei et al., 2022). See Figures 21, 22, 23 for the public test filtered versions.

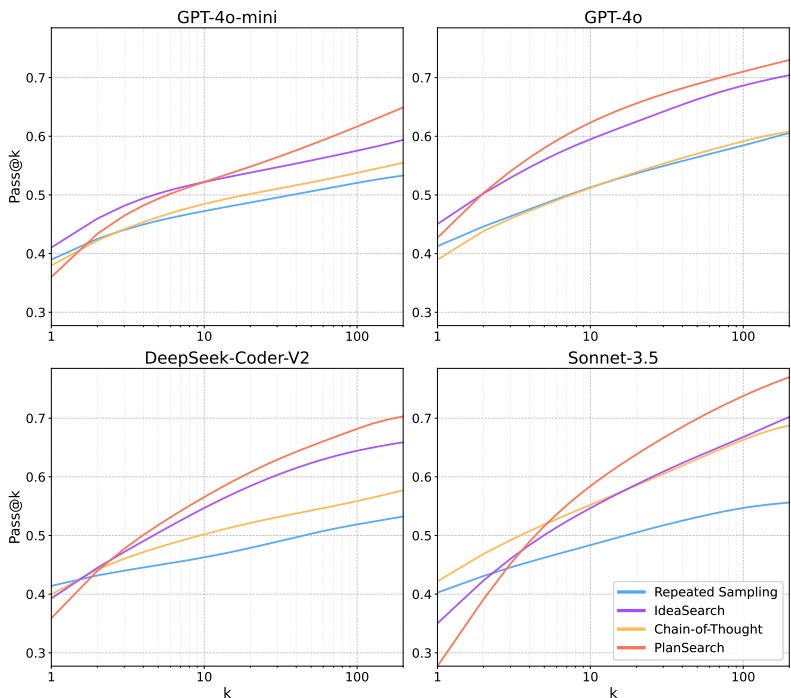

Figure 18: Pass@k graphs on LiveCodeBench, with the Chain-of-Thought baseline.

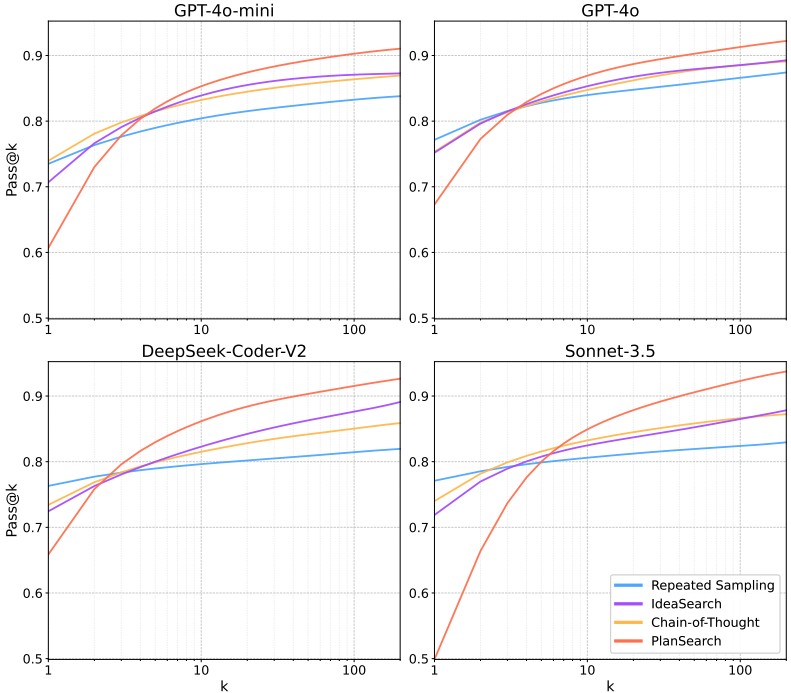

Figure 19: Pass@k graphs on MBPP+, with the Chain-of-Thought baseline.

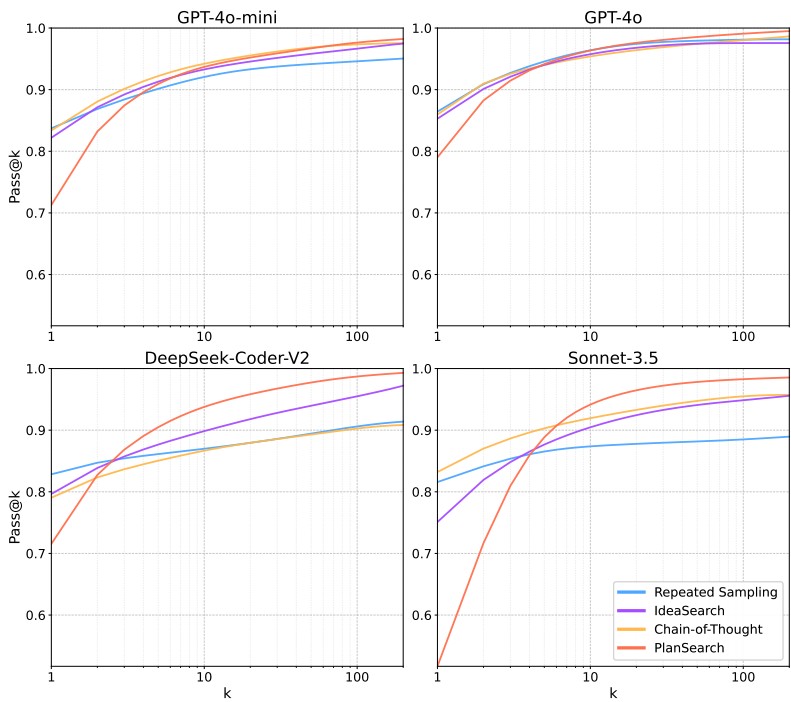

Figure 20: Pass@k graphs on HumanEval+, with the Chain-of-Thought baseline.

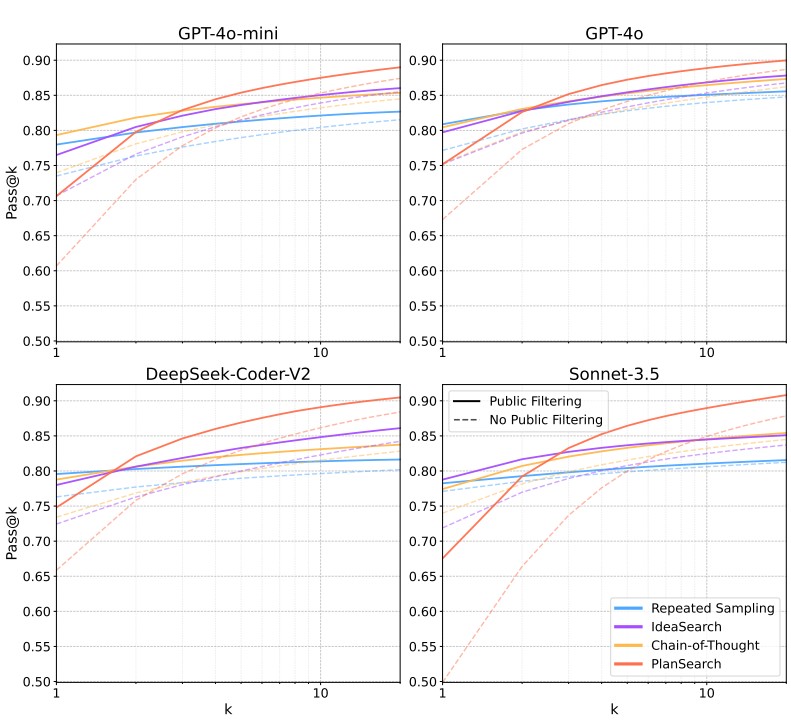

Figure 21: Pass@k graphs on LiveCodeBench, with the Chain-of-Thought baseline and public filtering.

Figure 22: Pass@k graphs on MBPP+, with the Chain-of-Thought baseline and public filtering.

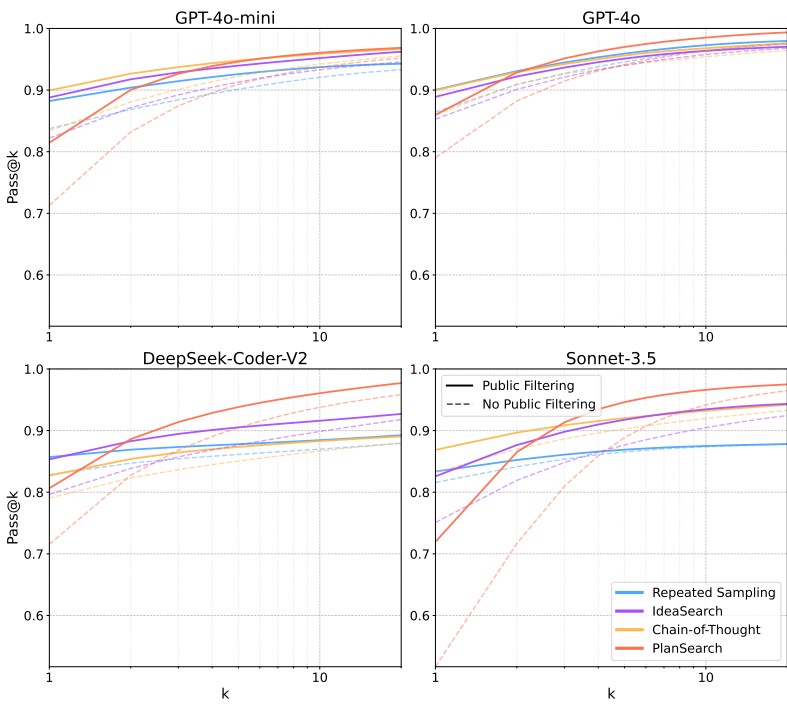

Figure 23: Pass@k graphs on HumanEval+, with the Chain-of-Thought baseline and public filtering.

# F ABLATION ON TEMPERATURE FOR REPEATED SAMPLING AND IDEASEARCH

See Figure 24. We sweep over temperature increments of $0.1$ from $0.0$ to $1.2$, inclusive, with top-$p$ of $0.95$, on REPEATED SAMPLING and IDEASEARCH.

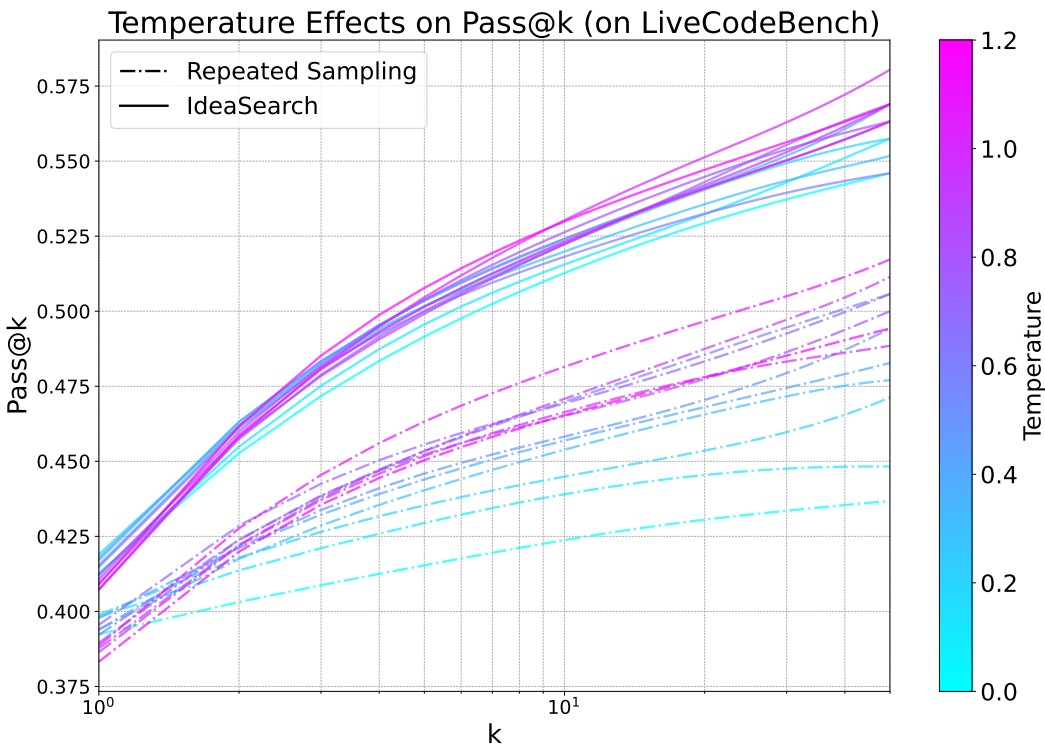

Figure 24: Sweep over temperature in $0.1$ increments from $0.0$ to $1.2$. REPEATED SAMPLING and IDEASEARCH both exhibit pass@k improvements at higher temperature, although it seems that higher temperatures may begin to plateau.

## G  DIVERSITY SCORE VS SEARCH IMPROVEMENT PLOTS FOR MBPP+ AND HUMANEVAL+

See Figures 25, 26, 5. Each figure is made through running the diversity measure as described in Section 6.1 on the generated codes of each run, then compared with the relative gain from pass@k compared to pass@1.

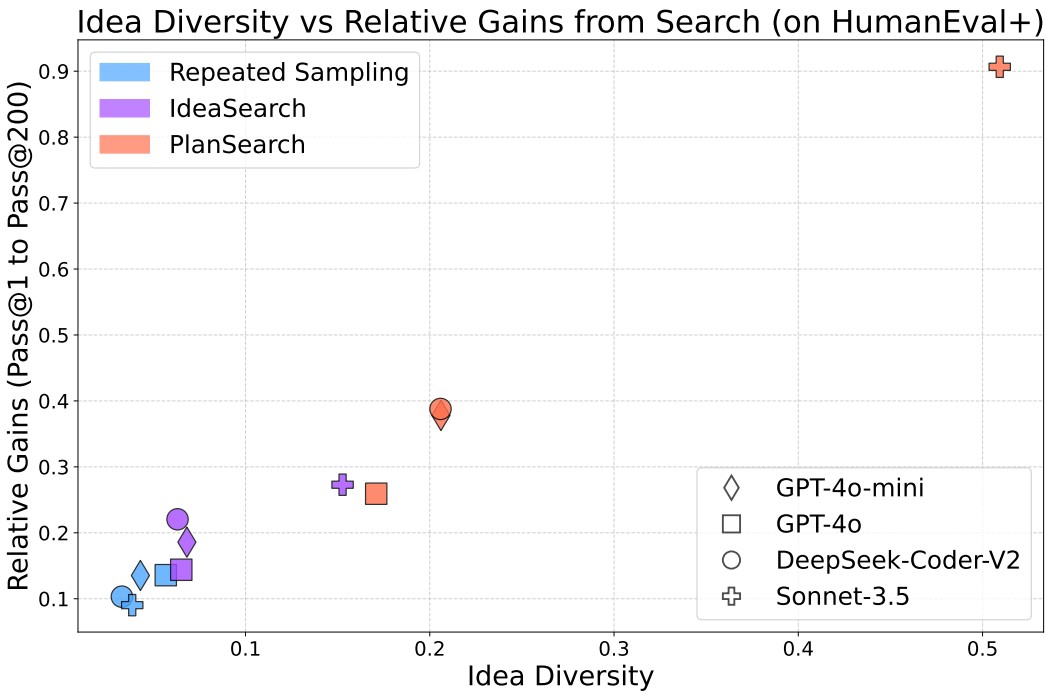

Figure 25: Relationship between the measured diversity score as described in Section 6.1 (where higher is more diverse) and relative improvement from the pass@1 of the method to the pass@200 of the method.

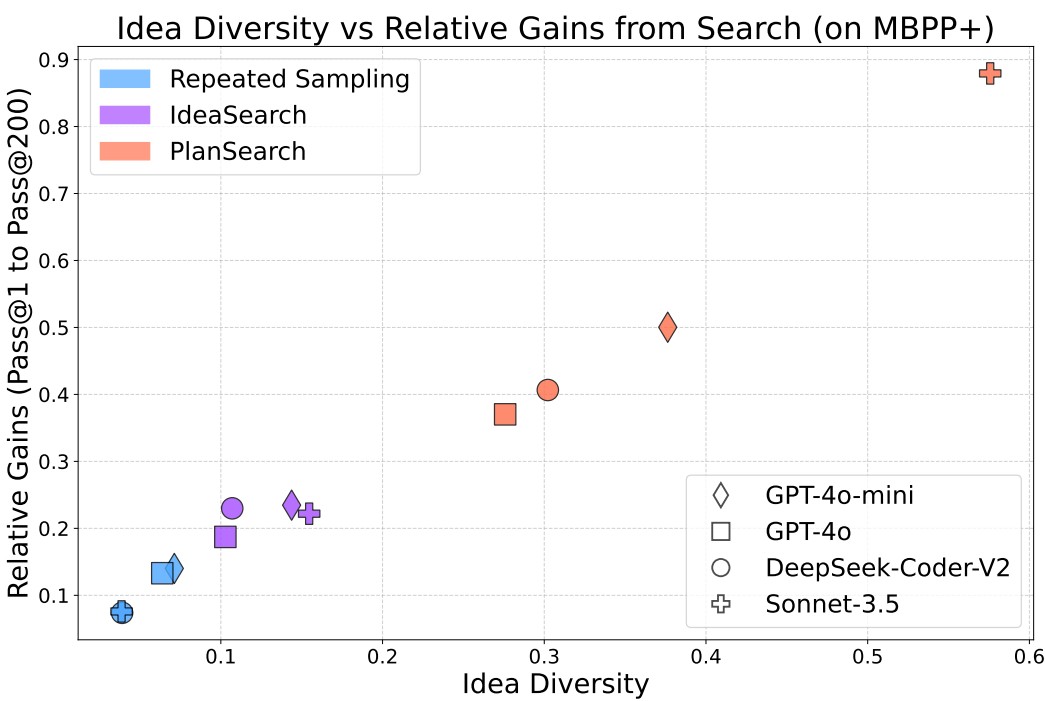

Figure 26: Relationship between the measured diversity score as described in Section 6.1 (where higher is more diverse) and relative improvement from the pass@1 of the method to the pass@200 of the method on MBPP+.

# H ABLATIONS

We run ablations on the core parts of PLANSEARCH, using GPT-4o-mini on LiveCodeBench (Jain et al., 2024). On the diverse observation generation side, we first verify our choice of $S = 2$—the maximum subset size to sample from out of a given pool of observations—and also compare performance across varying the number of observation layers used (Figures 27, 28). On the implementation side, we compare different sections of the proposed pipeline (see Figure 2) to translate combinations of observations to code in Figure 29. We compare each method's pass@k from $k = 1$ to $k = 200$.

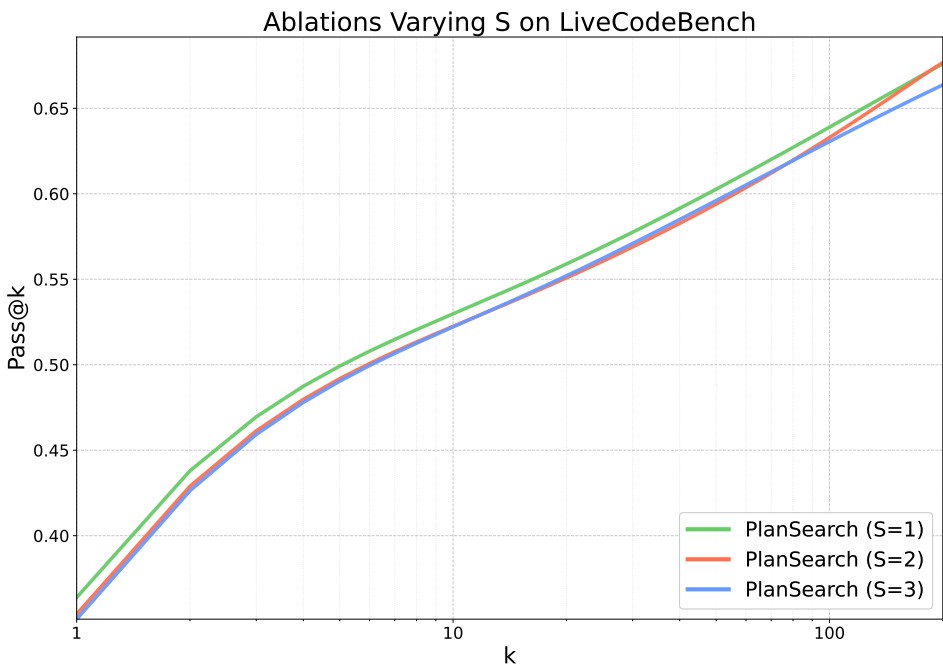

Figure 27: We run ablations of PLANSEARCH with different $S$—controlling the maximum subset size from a given pool of observations. There is not a large difference between different $S$, but we find that 1 or 2 is slightly more optimal, although if more completions are desired, $S$ can be increased to 3 as well.

The effect $S$, the maximum observation subset size to build upon, has on performance is not overly significant; there are small degradations as $S$ is increased to 3, but not noticeable. We choose $S = 2$ to obtain more code completions. See Figure 27.

Increasing $L$, the maximum number of layers of the observation tree, increases pass@k at large enough $k$ (above 50). We choose $L = 2$ to strike a balance between extracting a large pass@k gain while keeping compute costs reasonable. See Figure 28.

From Figure 29, we see that our overall translation step adds minor pass@k gains. We deconstruct the translation step into parts: the pseudocode step, the fix step (i.e., asking the model to fix its proposed solution sketch), and creating the solution sketch at all (which includes the fix step).

- "No solution sketch, no pseudocode" implies using a given observation combination to directly prompt for the solution code.

- "No pseudocode" implies skipping the pseudocode step. In other words, given a solution sketch, the sketch is directly translated into code.

- "No fix step, no pseudocode" implies the fix step is skipped, as well as the pseudocode. In order to have the same number of completions, the whole PLANSEARCH pipeline is run twice.

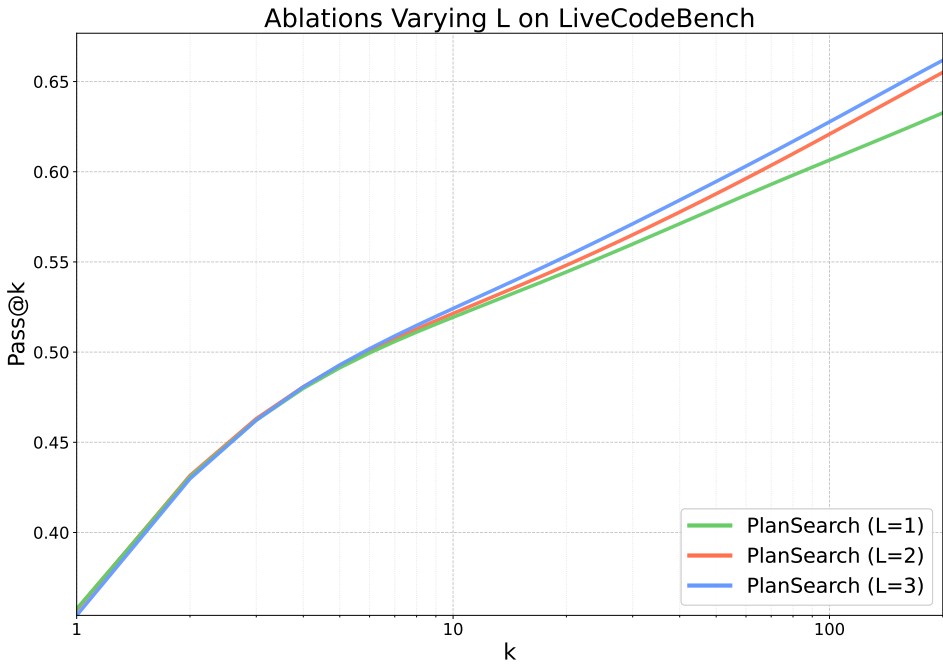

Figure 28: We run ablations of PLANSEARCH with different $L$—controlling the maximum order of observation used, i.e., how many layers the observation tree will search. We find pass@k scales with respect to increasing $L$

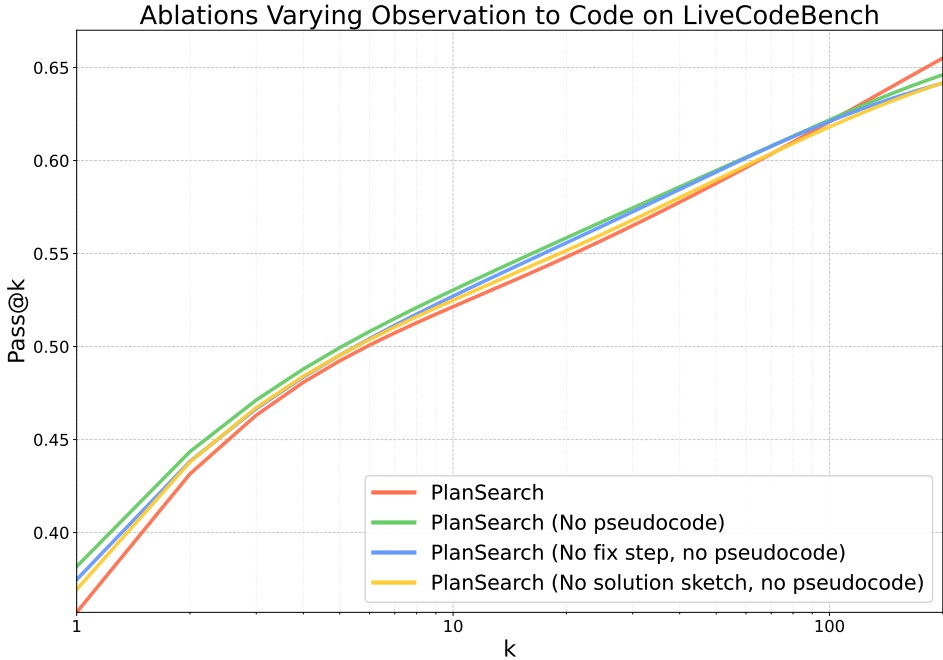

Figure 29: We run ablations of PLANSEARCH with different methods of translating a combination of observations to code.

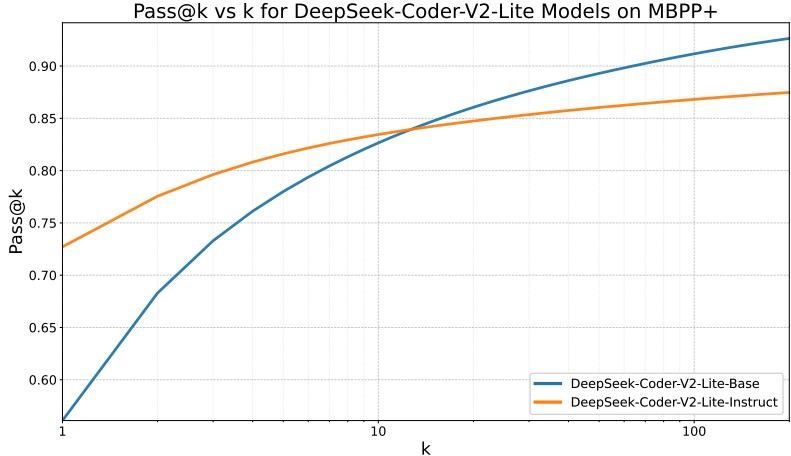

Figure 30: Despite DeepSeek-Coder-V2-Lite-Base having significantly lower pass@1 than its instruct counterpart, we observe that this trend reverses as $k$ increases, suggesting that the instruct model has less diversity than its base model counterpart. We observe this trend for many, but not all, models and benchmarks, and provide the full data in Appendix I.

## I    BASE MODELS VS. INSTRUCT MODELS FOR LARGE SAMPLES

We find that base models, despite performing poorly relative to their instruct counterparts for evaluated with pass@1, will frequently match or even exceed performance on pass@k for sufficiently high $k$. This is likely due to higher amounts of diversity in base models, which have not undergone post-training designed to elicit a single strong response from the model.

We see this effect across all models for HumanEval+ and MBPP+, but only the DeepSeek-Coder-V2 family for LiveCodeBench.

See Figure 31 for all base and instruct model comparisons between DeepSeek-Coder-V2-Lite, Llama-3.1-8B, and Llama-3.1-70B on all three datasets.

We also provide Llama-3.1-8b and DeepSeek-Coder-V2-Lite pass@k comparisons for $k$ up to $10,000$; see Figures 33, 32.

## J    BASE MODELS VS. INSTRUCT MODELS WITH PUBLIC TEST FILTERING

We repeat the graphs from Appendix I, but with public test filtering. We find that base models with public test filtering almost always exceed the pass@1 of their instruct model variants.

See Figure 34 for all base and instruct model comparisons between DeepSeek-Coder-V2-Lite, Llama-3.1-8B, and Llama-3.1-70B on all three datasets with public test filtering.

We also report Llama-3.1-8b and DeepSeek-Coder-V2-Lite pass@k comparisons with public test filtering for $k$ up to $10,000$; see Figures 35, 36.

Pass@k of Base and Instruct Models

Figure 31: Pass@k curves comparing DeepSeek-Coder-V2-Lite, Llama-3.1-8B, and Llama-3.1-70B base and instruct performance.

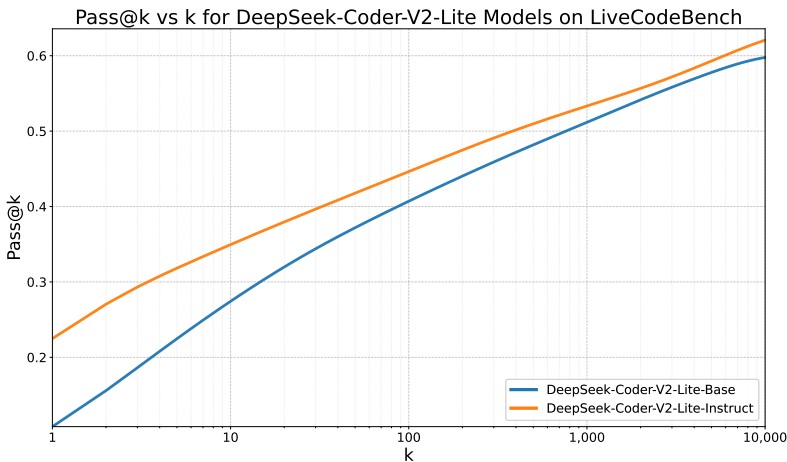

Figure 32: Pass@k curves comparing DeepSeek-Coder-V2-Lite's base and instruct versions on LiveCodeBench with up to $10,000$ completions.

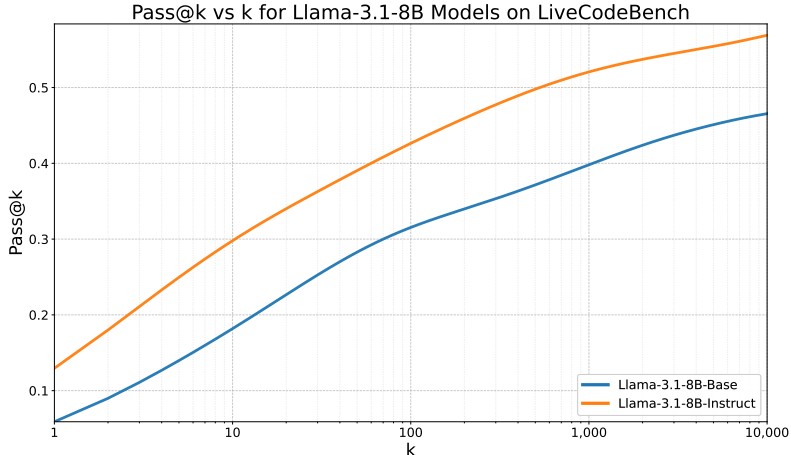

Figure 33: Pass@k curves comparing Llama-3.1-8B's base and instruct versions on LiveCodeBench with up to $10,000$ completions.

Figure 34: Pass@k curves comparing DeepSeek-Coder-V2-Lite, Llama-3.1-8B, and Llama-3.1-70B base and instruct performance with public test filtering.

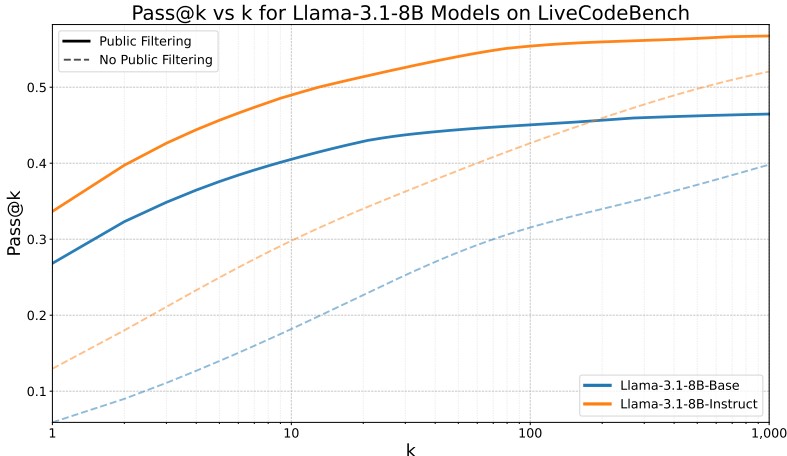

Figure 35: Pass@k curves comparing Llama-3.1-8B's base and instruct versions on LiveCodeBench with up to $1,000$ completions and with public test filtering.

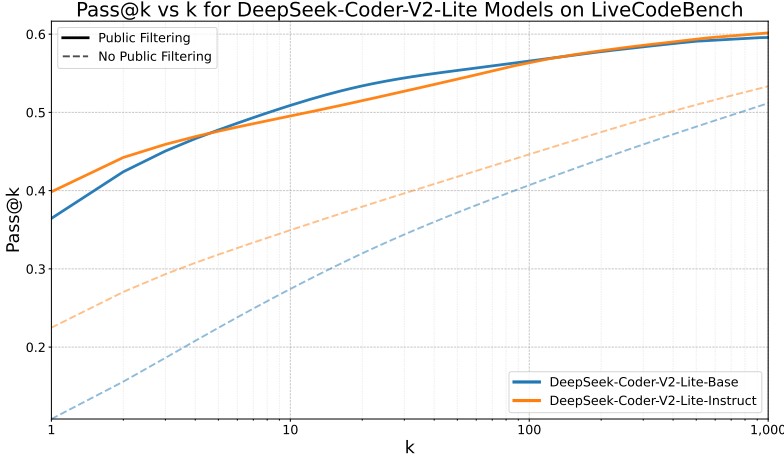

Figure 36: Pass@k curves comparing DeepSeek-Coder-V2-Lite's base and instruct versions on LiveCodeBench with up to $1,000$ completions and with public test filtering.

# K  OPENAI O1

We also do a brief analysis of OpenAI's o1 (OpenAI, 2024) models. We run PLANSEARCH on top of o1-mini, which is the best o1 model available through API access at the time of writing for competitive coding. Due to both severe cost constraints and dataset saturation on HumanEval+ and MBPP+, we only run o1-mini on LiveCodeBench (Jain et al., 2024).

First, we investigate its pass@k trends on LiveCodeBench. (See Figure 37.) As expected, PLANSEARCH trails REPEATED SAMPLING for low $k$. However, at high $k$, PLANSEARCH competes with/slightly outperforms REPEATED SAMPLING.

We believe there to be two reasons why there is not as large of an improvement compared to other non-search-based models. First, there may be diminishing returns when stacking multiple search methods on top of each other naïvely. Second, LiveCodeBench is noticeably saturated, reaching solve-rates of over $90\%$. Thus, it is much harder to notice large jumps in dataset performance, as can be seen in pass@k curves (see Figure 6) on HumanEval+ (Liu et al., 2023). There, REPEATED SAMPLING, IDEASEARCH, and PLANSEARCH all perform roughly the same, with PLANSEARCH only slightly outperforming the others. We believe a combination of these effects also implies a greater $k$ is required to notice a large difference between REPEATED SAMPLING and PLANSEARCH.

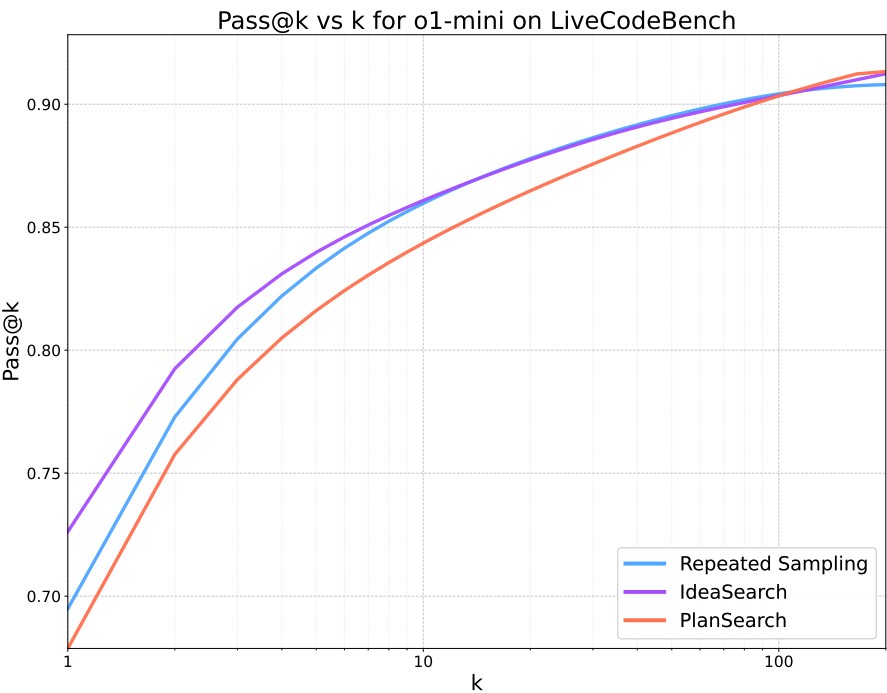

Figure 37: Pass@k performance methods applied on top of o1-mini, on LiveCodeBench, plotted over $k \in \{1, \ldots, 200\}$. PLANSEARCH slightly outperforms REPEATED SAMPLING at high $k$; larger $k$ is not run due to cost constraints.

We notice that the OpenAI frequently refuses to answer queries from PLANSEARCH, likely due to efforts to minimize revealing the full chain-of-thought the model is using. To combat this, we find that filtering out the words `steps`, `step`, and `quote` reduces the fraction of unanswered queries from roughly $40\%$ to less than $0.5\%$. For any query that is still refused after this approach, we route the same prompt to GPT-4o. In addition, we remove the pseudocode step to reduce the amount of flagged responses.

On the diversity end, we apply the same methodology as described in Section 6.1. The corresponding plot can be found in Figure 5. Even though o1-mini is strong on its own, we find that it is not very diverse, which is consistent with our claim that diversity correlates with relative performance gain seen with increasing $k$.

## L  Bar Charts

See Figures 38, 39, 1. These plot pass@1 and pass@200 of select methods between REPEATED SAMPLING and PLANSEARCH, on datasets MBPP+, HumanEval+ (Liu et al., 2023), and Live-CodeBench (Jain et al., 2024).

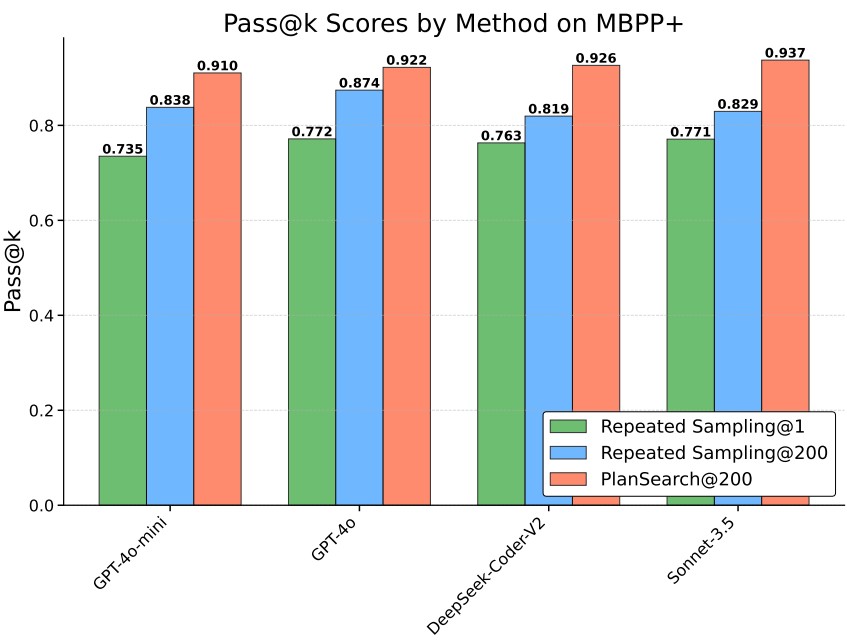

Figure 38: Bar chart with REPEATED SAMPLING@1, REPEATED SAMPLING@200, and PLANSEARCH@200, on MBPP+.

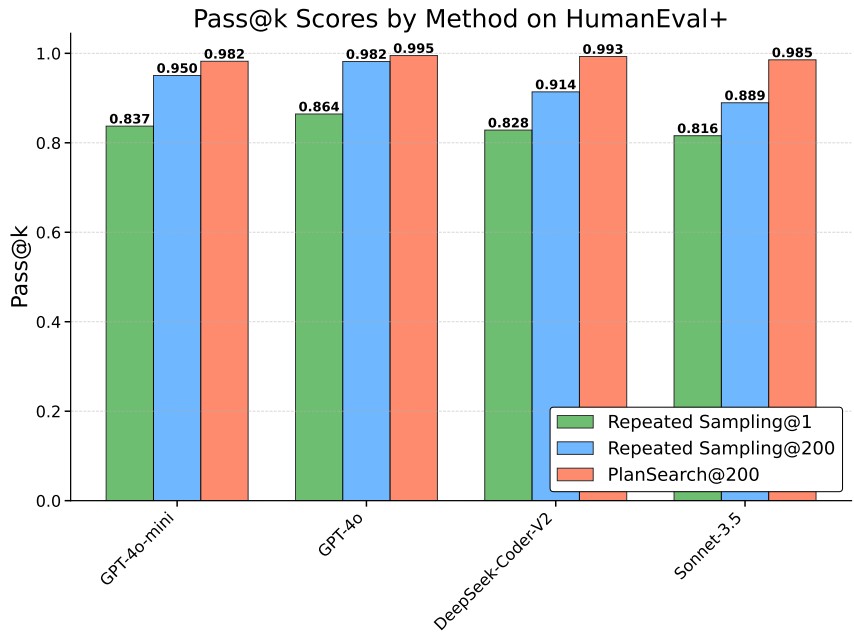

Figure 39: Bar chart with REPEATED SAMPLING@1, REPEATED SAMPLING@200, and PLANSEARCH@200, on HumanEval+.

# M PROMPTS

## M.1 BACKTRANSLATION

### M.1.1 BACKTRANSLATE SYSTEM PROMPT

```
You are an expert Python programmer. You will be given an
algorithmic question (problem specification). You will return
a high-level, natural language solution to the question, like
an editorial. You will NOT return any code. Be as creative as
possible, going beyond what you think is intuitively correct.
```

### M.1.2 IMPLEMENT BACKTRANSLATION IDEA

```
You are an expert Python programmer. You will be given a question
(problem specification) and a natural language solution/tutorial
that describes how to solve the problem. You will generate
a correct Python program that matches said specification and
tutorial and passes all tests. You will NOT return anything
except for the program inside markdown codeblocks.
```

## M.2 REPEATED SAMPLING

```
You are an expert Python programmer. You will be given a question
(problem specification) and will generate a correct Python program
that matches the specification and passes all tests. You will NOT
return anything except for the program inside Markdown codeblocks.
```

## M.3 SIMPLE IDEA

```
You will given a competitive programming problem; please output
a high-level description of how to solve the problem in natural
language. Below are examples:

Example input: PROBLEM DESCRIPTION HERE
Example output: EXAMPLE OUTPUT HERE
Here is the competitive programming problem: PROBLEM TO SOLVE

Brainstorm a high-level, natural language solution to the problem
above. Note that your intuition may lead you astray, so come up
with simple, creative ideas that go beyond what you would usually
come up with and go beyond your narrow intuition. Brainstorming
solutions that do not seem intuitively correct IS CRUCIAL.
```

## M.4 PLANSEARCH

### M.4.1 PROMPT FOR OBSERVATION PART 1

```
You are an expert Python programmer. You will be given an
competitive programming question (problem specification). You
will return several useful, non-obvious, and correct observations
about the problem, like hints to solve the problem. You will NOT
return any code. Be as creative as possible, going beyond what
you think is intuitively correct.
```

### M.4.2 PROMPT FOR OBSERVATION PART 2

```
You are an expert Python programmer. You will be given an
competitive programming question (problem specification) and
```

```
several correct observations about the problem.

You will brainstorm several new, useful, and correct observations
about the problem, derived from the given observations.  You will
NOT return any code.  Be as creative as possible, going beyond
what you think is intuitively correct.
```

### M.4.3 COMBINING OBSERVATIONS

```
Here is a sample prompt from the function with placeholders:

Here is the competitive programming problem:

Problem statement placeholder

Here are the intelligent observations to help solve the problem:

Observation 1 placeholder
Observation 2 placeholder
Observation 3 placeholder

Use these observations above to brainstorm a natural language
solution to the problem above.  Note that your intuition may
lead you astray, so come up with simple, creative ideas that go
beyond what you would usually come up with and exceeds your narrow
intuition.
Quote relevant parts of the observations EXACTLY before each step
of the solution.  QUOTING IS CRUCIAL.
```

## N   A MODEL OF REPEATED SAMPLING: PASS@K

Consider a simplified model of repeated sampling for code generation. Suppose we have a dataset $D = \{P_1, \ldots, P_l\}$ with $l$ problems. For some problem $P_i$, define the probability $p_i$ as the probability that our code generation model solves the problem $P_i$ in one submission. The pass@k (Chen et al., 2021; Kulal et al., 2019) metric (for problem $P_i$) is defined as the probability that our code generation model solves the problem $P_i$ at least once out of $k$ submissions. Thus, if we know the true $p_i$ of our model, we may compute our pass@k simply:

$$\text{pass@k}_i = 1 - (1 - p_i)^k \tag{2}$$

$$\text{pass@k} = \sum_i \text{pass@k}_i / l \tag{3}$$

However, it turns out that for $k > 1$, the naïve estimator as seen in Equation 2 is biased, if we sample $n_i \geq k$ from our code model to solve $P_i$, $c_i \leq n_i$ are correct, and compute $p_i = c_i / n_i$ (Chen et al., 2021). Instead, $\text{pass@k}_i$ is typically computed using the unbiased estimator:

$$\text{pass@k}_i = 1 - \frac{\binom{n-c}{k}}{\binom{n}{k}} \tag{4}$$

Note that reporting pass@k on a dataset where $l = 1$ is rather pointless, since pass@k can be derived using only $\text{pass@1}_1$ and $n_1$. Every curve, over a suitable range of $k$ values, will look like the *S-curve* seen in Figure 40 (as $k$ is plotted on a $\log$ scale).

However, with datasets where $l > 1$, models are able to differentiate themselves through larger $k$, since the overall pass@k is an average of these $l$ curves. For example, for $l = 3$, it is less optimal to have solved probabilities of $\text{Set1} = \{0.001, 0.7, 0.9\}$ versus $\text{Set2} = \{0.05, 0.1, 0.25\}$, in the regime of roughly $k = 20$ to $k = 2,000$ (in which both converge to 1), even though $\text{Set1}$ has a pass@1 of 53% and $\text{Set2}$ has a pass@1 of 13%. See Figure 41.

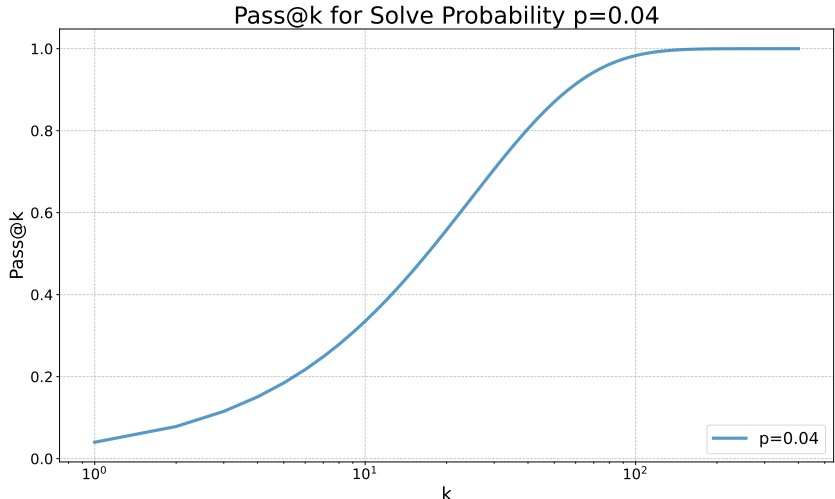

Figure 40: A simple pass@k '*S-curve*' plotted with $1 - (1 - p)^k$, where $p = 0.04$.

Although not shown in the graph, $\mathrm{Set2}$ converges close to $1$ at roughly $k = 400$, several orders of magnitude below $\mathrm{Set1}$. In addition, note that the slight notch seen in $\mathrm{Set1}$'s curve at large $k$ is due to the presence of low, but non-zero solve-rates, which can be seen in empirical pass@k curves later on. (These can be thought as the beginning of the 'ramping-up' regime of the typical *S-curves* in Figure 40.)

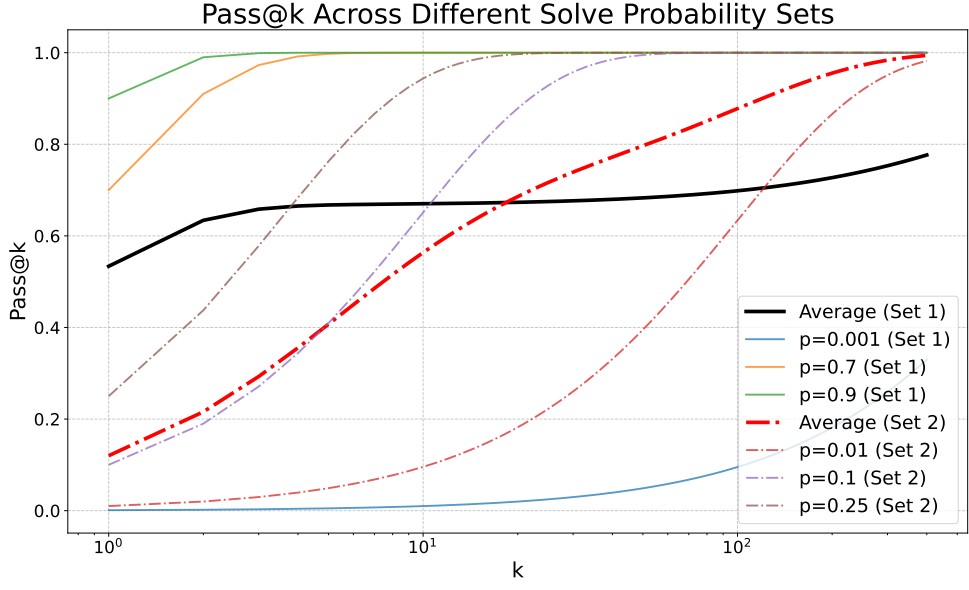

Figure 41: Two pass@k curves on a hypothetical dataset of length $l = 3$, and the solve probabilities of Set 1 are $\{0.001, 0.7, 0.9\}$ and Set 2 are $\{0.05, 0.1, 0.25\}$. Note that the pass@1 is $53\%$ and $13\%$, respectively. However, at roughly $k = 20$, Set 2 surpasses Set 1 and within an order of magnitude, achieves pass@k of roughly $1.0$.

## O    BIASED ESTIMATOR FOR PASS@K DUE TO NON-INDEPENDENCE OF PLANSEARCH

From a pure theoretical standpoint, the expression is biased (if using the same interpretation), but it still leads to a similar interpretation—computing the probability that a subset of size $k$ drawn from the set of samples we already generated contains at least one success. (These given samples were generated by one run of PLANSEARCH.) As such, in theory, the estimator may be slightly biased in the PLANSEARCH case when computing its true pass@k. In practice, we do not believe this to be a large concern, especially as our primary results feature a relatively large $k = 200$.

### O.1    MEASURING DIVERSITY

```
You are an expert Python programmer.  You will be given a
competitive programming problem and two pieces of code which
are attempts to solve the problem.  For your convenience, you
will also be given the idea for each code, summarized in natural
language.  You will be asked to answer whether the ideas behind
the code are the same.  You must ONLY output 'Yes.'  or 'No.'
```

## P    COMPETITIVE PROGRAMMING

Competitive programming is a popular subset of programming tasks that involve solving complex algorithmic reasoning. Typically, problems consist of a problem statement (written in natural language) $P$, with associated tests: $(x_i, y_i), i \in \{1, \dots, m\}$, for which any solution must pass all of them.

The number of tests $m$ depends on the problem, but typically ranges on the order of 25 to 100. A small subset of the tests are typically given to the solver (we call these public tests) to use as validation that their program passes simple cases. The rest of the tests are hidden. Solutions to the problems must generally pass all the tests to be considered correct. Formally, we let $f(x)$ denote the output of said code ran on input $x$. The solution code is considered correct (passing) if and only if $f(x_i) = y_i$ for all $i \in \{1, \dots, m\}$.

Each dataset consists of many (on the order of low-hundreds) independent problems, and models are evaluated on each of these problems independently.

## Q    OTHER RELATED WORK

### Q.1    SEARCH IN CLASSICAL AI

Classical search algorithms like breadth-first search, depth-first search, and A* search have been widely used for pathfinding, planning, and optimization (Russell & Norvig, 2002). More advanced search techniques like Monte Carlo Tree Search (MCTS) have achieved remarkable success in domains like game playing, enabling superhuman performance in Go (Silver et al., 2016; 2017), Poker (Brown & Sandholm, 2018; 2019) and Diplomacy (FAIR et al., 2022). More recently, scaling laws have been found for the performance of AI systems in board games, where ELO improves logarithmically with the amount of compute spent at inference (Jones, 2021).

### Q.2    SEARCH WITH LANGUAGE MODELS

Applying search on top of LLMs has been a topic of much interest, especially with an eye towards code generation (Chen et al., 2021; Li et al., 2022). Historically, methods such as beam search significantly improved performance for translation systems (Freitag & Al-Onaizan, 2017). Closer to the present day, several recent works have explored repeated sampling (Chen et al., 2024; Brown et al., 2024; Bansal et al., 2024; Wu et al., 2024) as a search method for improving performance. Repeated sampling is a method which directly generates candidate code solutions from the model many times at moderate to high temperatures in hopes that one of the resulting generations will be correct. However, although these works address the roughly linear increase in pass@k with respect

to $\log k$, they only focus on the most basic version of repeated sampling, without searching in idea space.

When combined with a verifier, reward model, or other filtering algorithm to select the best generation (in cases where pass@k is not a viable metric due to lack of test cases), it is also known under the name of best-of-n sampling (Mudgal et al., 2024). Many works show somewhat good results under intelligent selection of such a filtering algorithm (Chen et al., 2022a; 2024). Recently, several approaches have demonstrated the power of repeated sampling. For example, repeated sampling from a small model can sometimes outperform taking a single sample from a large model on an equalized compute bases (Snell et al., 2024). Unlike algorithms such as repeated sampling, which search over the output space, the key insight of PLANSEARCH is that it is far more effective to instead search plans over the *latent idea space*. By explicitly searching over different natural language plans before generating the code, we significantly increase the diversity of the final code outputs and thus, the resulting pass@k scores for sufficiently large k.

## R    PUBLIC TEST FILTERING

Public test filtering is a method which only chooses samples out of the original pool $n$ which pass the public tests. This is particularly useful in settings such as code deployment where executing the full suite of tests may be computationally costly or otherwise undesirable (e.g. in a coding contest where every incorrect submission is penalized). Thus, instead of submitting all $n$ codes, after public test filtering, only codes $c_i$ would be submitted such that $c_i(x_j) = y_j$ for all $j \in \{1, \ldots, u\}$, where $c_i(x)$ refers to the output from running the code on some input $x$. The primary effect of public test filtering is to shift the pass@k curve leftward, since public test filtering will discard low quality candidate solutions that either fail to compile or fail elementary test cases for the problem.

All problems in MBPP+, HumanEval+, and LiveCodeBench come with a few public tests which are usually used to sanity check any submissions. We can further improve performance by filtering on these public tests before a final submission, as described. Applying public test filtering reduces the number of samples to achieve the same accuracy by tenfold: PLANSEARCH to achieve a 77.1% accuracy on LiveCodeBench after just 20 submissions (pass@20) compared to a pass@200 of 77.0% without using public filtering (see Figure 4). We provide full results for the other datasets in Appendix B.

## S    MATHEMATICS AND EXAMPLES OF THE DIVERSITY MEASURE

While our choice of a diversity metric is intuitive, one should note that there are a number of intriguing details that result from our definition.

For example, *it is* the case that with $k$ unique ideas and $n$ samples of each idea, respectively (for a total of $kn$ total generated codes), we achieve a diversity score approaching $(k-1)/k$.

For a quick proof, suppose that there are $k$ cliques, each of $n$ size. Each clique represents a unique idea. We wish to capture the number of unfilled edges over the number of possible edges as our diversity score:

$$\frac{\binom{k}{2}n^2}{\binom{kn}{2}} = \frac{(k-1)n}{kn-1} \tag{5}$$

which converges to $\frac{k-1}{k}$ as $n$ grows large.

Our formulation seems more intuitive than other proposals, such as one that simply counts how many unique ideas $k$ lie within a pool of size $n$ to compute $k/n$ as the diversity score.

For instance, suppose there are only two unique ideas, one clique of which is of size $2n - 1$, and the other only output once. The simple proposal would compute both the diversity score of this uneven group *and* that of an even group (where both ideas are output $n$ times) to be $\frac{2}{2n} = \frac{1}{n}$.

However, our score would compute the even group to be $\frac{n}{2n-1}$, and the uneven group as:

$$\frac{1}{2n} \cdot 1 + \frac{2n-1}{2n} \cdot \frac{1}{2n-1} = \frac{1}{n} \tag{6}$$

Instead of counting edges, we compute the probability that two randomly selected outputs have similar idea to each other, which is another interpretation of our diversity score.

It seems clear that a case with $2n - 1$ instances of idea 1 and 1 instance of idea 2 is 'less diverse' than a case with $n$ instances of both idea 1 and idea 2. A naïve proposal may score these two as being the same diversity, whereas our score scores them as $1/n$ and roughly $1/2$, respectively.

## T    LIMITATIONS AND FUTURE WORK

While PLANSEARCH substantially improves diversity over idea space at inference-time, fundamentally, improvements in diversity should come at the post-training stage, like with methods such as o1 OpenAI (2024). This likely requires re-imagining the post-training pipeline for LLMs around search, instead of the current paradigm optimized for a single correct response. We are optimistic about future work in designing improved post-training objectives to maximize both quality and diversity, while specifically optimized to use inference-time compute to maximum effectiveness.

PLANSEARCH and IDEASEARCH tradeoff a slight deterioration of pass@1 performance for a large improvement in pass@k performance. However, in many such cases outside of code generation, it is infeasible to run an LLM-based model for more than a few attempts at most. For example, in Figure 8, PLANSEARCH does not significantly outperform REPEATED SAMPLING until $k \geq 4$.

Fortunately, many filtering algorithms exist, which mitigates this tradeoff by implicitly bringing pass@k (for high $k$) to pass@1 (or lower $k$), i.e. shifting the original pass@k curve leftward. Even the simplest filtering—public test filtering—improves PLANSEARCH's pass@1 significantly above REPEATED SAMPLING's pass@1, which continues as $k$ increases. Moreover, *most to almost all* base models with public test filtering outperform their instruct model variants at pass@1, no matter the dataset (see Appendix J). Since base models' pass@1 is known to be worse than instruct models to trade off for higher diversity, we suggest a new paradigm—developing search algorithms which tradeoff pass@1 performance for much stronger pass@k performance, then filtering the generated solutions to extract the pass@k *back into* pass@1.

With good filtering methods, which we demonstrate can be simple in nature, pass@k, for medium $k$, can be effectively brought down to pass@1, emphasizing a similar paradigm of increasing diversity, then strengthening existing filtering methods, even for domains outside of code generation that are out of scope of this paper.

A natural extension of this work is training the underlying model itself on successful plans and code solutions obtained from PLANSEARCH. This has the potential to distill the pass@k into the pass@1—without inference-time methods like filtering—by reducing the likelihood of the model going down unfavorable branches of the search tree. We believe that such training is likely to significantly improve the model and look forward to future work in this direction.

In terms of methodological improvements to PLANSEARCH, PLANSEARCH currently searches all leaf nodes in the search tree uniformly. Because of this, it becomes quickly intractable to go further than a few levels deep, and in our experiments, we are only able to go two levels down the tree. Several approaches based on Monte-Carlo Tree Search (MCTS), such as Tree of Thought Yao et al. (2023a) or Reasoning as Planning (Hao et al., 2023a), have suggested that some form of dynamic pruning and expansion of nodes can be very helpful. We are optimistic that PLANSEARCH can be further improved by such methods.

Furthermore, PLANSEARCH is a fairly elementary method taking advantage of the paradigm that searching over a *conceptual or idea space* is an effective method to improve diversity, and thus, downstream task performance. It is completely feasible to search at an even higher level of abstraction than observations, which may be used to inject even more diversity into the final generated outputs.

## U    COMPARISON WITH AGENTIC MODELS

We also run baselines with a basic 'agentic' model which is allowed a code execution environment with the given public tests. To do this, it first generates code similar to REPEATED SAMPLING, then runs said code on the given public tests. If the code does not pass, the error is returned to the model

and the model is prompted to fix the code. This process continues until either the model successfully passes the public tests or it reaches a certain iteration limit $T$.

We compare such a model with our baselines as well. For all comparisons, we set $T = 10$. Note that the agentic models do have access to a code execution environment, which is an inherently different paradigm than all of our baselines. Thus, we do two comparisons.

We first compare the *last* submission of the agentic model (which is either passing public tests or does not complete in the allotted iteration limit) versus the public-filtered versions of each of our baselines. The rationale behind this is that in both methods, all the faulty submissions which do not pass public tests are discarded, and both have access to code execution. The results are seen in Figure 42.

Figure 42: Pass@k curves comparing the basic agentic model's last submissions only on Live-CodeBench with up to 20 completions and public test filtering.

Next, instead of stopping when successful, the agentic model continues to iterate even when the code is successful. If the code is successful and uses $i$ iterations out of $T$, the agentic model starts over from scratch, except with $T - i$ total iterations instead. We compare *all* submissions of this agentic model versus the default versions of each of our baselines, since the submissions of both methods are not guaranteed to pass public tests. Note that this setup is still slightly unfair for our baselines, since the last submissions of the agentic model will have already iterated and extracted sufficient signal from public tests, whereas our baselines do not use any test execution signal at all. The results can be found in Figure 43.

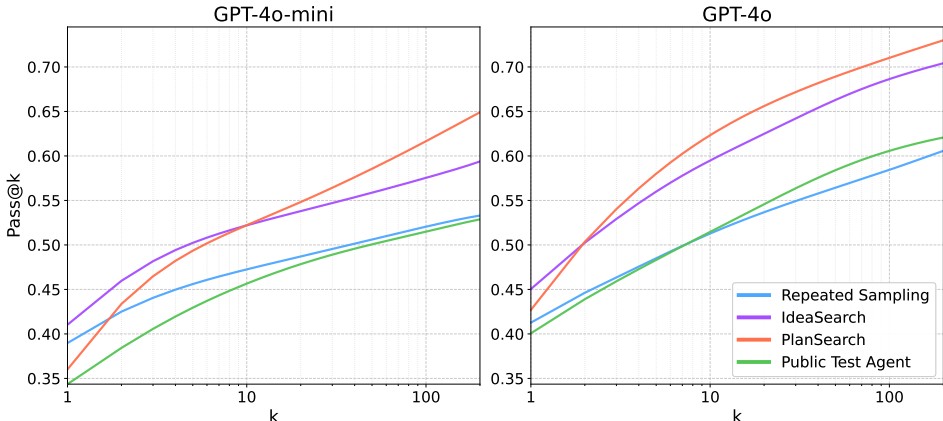

Figure 43: Pass@k curves comparing the basic agentic model with all $T$ generations on Live-CodeBench with up to 200 completions.

