# OpenReview forum: "Planning in Natural Language Improves LLM Search for Code Generation"
_ICLR.cc/2025/Conference — ICLR 2025 Spotlight_

### Official Review · Reviewer_KeqM · 2024-10-31

**Soundness:** 3
**Presentation:** 3
**Contribution:** 3
**Rating:** 6
**Confidence:** 3

**Summary:**

The paper proposes a new methodology to improve language model test time scaling efficiency. Their study shows that simply sampling more outputs from the same input prompt will have limited diversity, thus limiting the improvement of the model performance. To avoid this issue, they propose to sample the natural language plan of the solution first and then ask the language model to generate a solution based on the plan. They propose an effective method for sampling various plans, and experiments show that the proposed method improved the model accuracy in best-of-n settings.

**Strengths:**

The paper is well-written. Analysis and ideas are presented progressively, making it easier to follow. The proposed method is well-motivated by the polarized results of different solution sketches. The experiment results also show clear improvement over the baseline model and methods.

**Weaknesses:**

1. Although the method shows promising results with models including Claude 3.5 sonnet and GPT-4o, etc., the improvement over o1-mini is marginal. Does this suggest the method is not compatible with other inference-time scaling compute?
2. While the proposed method shows promising improvement compared to naive sampling and Pass@1, how does it compare to another search-based method, like MCTS?

**Questions:**

Please refer to the weakness.

---

> ### Author Response · Authors · 2024-11-23
> **Response to Reviewer 3**
>
> Thank you for your review. We are glad you found the paper clear, well-motivated, and appreciated the progression of ideas and experimental results. Below, we hope to address your concerns.
>
> # o1-mini
>
> > Although the method shows promising results with models including Claude 3.5 sonnet and GPT-4o, etc., the improvement over o1-mini is marginal. Does this suggest the method is not compatible with other inference-time scaling compute?
>
> Thank you for your observation. Our Appendix K describes our perspective on this observation, which we expand on here. Indeed, the gains on the test split of LiveCodeBench with o1-mini are somewhat marginal, but we believe this is in line with the main claims of our paper. The reasons are threefold:
>
> - Firstly, our algorithm was not explicitly designed for search on top of existing search algorithms, and we conducted almost all our experiments before models such as o1 were announced. The purpose of our paper is mainly to show that using inference-time compute on top of standard post-trained models can be extremely effective, by adding much needed model diversity.
> - o1-mini is already nearly saturated on LiveCodeBench. The pass@200 of o1-mini with naive repeated sampling is already hovering close to 0.9. Any additional method on top of an already-saturated method will likely show limited absolute improvements compared to unsaturated tasks. For example, we see analogous patterns when applying PlanSearch on models like GPT-4o on HumanEval+, which is notably saturated. In Figure 6 (top-right), PlanSearch on GPT-4o marginally improves over IdeaSearch and Repeated Sampling. However, on *harder, unsaturated* datasets such as LiveCodeBench (for non-o1 models), **the pass@k of Repeated Sampling and PlanSearch clearly diverges again** as k becomes large. Thus, with an even harder, unsaturated task, PlanSearch with o1-mini may exhibit large differences as well.
> - Lastly, as seen in Figure 5, o1-mini is **noticeably less diverse than all other models**. (This may be due to the specific post-training algorithms o1-mini was trained with, referring back to the conversation with Reviewer 2 on lack of diversity in common post-training algorithms, or may be due to saturation on the dataset.) No matter the reason for this lack of diversity, it is true (as seen in all our diversity plots) that idea diversity is strongly correlated with relative gains from search. This correlation implies that models with lower diversity are predicted to exhibit less significant pass@k boosts, even with PlanSearch.
>
> # MCTS and Other Methods
>
> > While the proposed method shows promising improvement compared to naive sampling and Pass@1, how does it compare to another search-based method, like MCTS?
>
> As far as we know, there is no well-established MCTS baseline for competitive coding and there are several reasons for this. Doing MCTS over token space would likely have nodes representing the current code prefix, with each step adding a new code token to the end of the prefix, and exploring only promising code prefixes through some value function. If the value function is some reasonable function of the LLM logprobs, this whole algorithm could be thought of as a sophisticated LLM sampling algorithm. The effect of this would only be to filter some generated code outputs from Repeated Sampling, which would not get rid of the inherent diversity problem in Repeated Sampling and may possibly even lose diversity given emphasis of the shared code prefixes.
>
> However, suppose the MCTS is run instead on idea space. Viewing this algorithm at a high level, we see that the algorithm may be regarded as a derivative of PlanSearch, instantiated as a filtered tree search over ideas. Instead of going through all the brute-force combinations and rolling out every branch, an MCTS-like algorithm would filter the most promising branches to go down. We consider this to be an exciting future direction for improving PlanSearch and describe this and other variations in the Limitations section of our paper.
>
> For other baselines, we show comparisons with an agentic model in our response above to Reviewer 2. We hope these responses address your questions and provide clarity on the points you raised. Thank you again for your thoughtful feedback, and we welcome any further discussion.

---

### Official Review · Reviewer_CksG · 2024-11-02

**Soundness:** 3
**Presentation:** 3
**Contribution:** 2
**Rating:** 6
**Confidence:** 3

**Summary:**

The authors describe a new method for code generation (for short problems). It involves generating a set of observations about the problem (in natural language), constructing plans to solve it and then generating solutions.

**Strengths:**

1. The authors achieve impressive results on three benchmarks, e.g. they make Claude 3.5 Sonnet achieves a pass@200 of 77.0% on LiveCodeBench.
2. They provide an interesting motivation for their work: showing that idea generation, not idea execution is the bottleneck of LLMs and they the solutions they generate lack diversity. It's interesting that "a correct sketch is sufficient to produce the correct final solution with relatively high accuracy, even only after 10 tokens of backtranslated solution." and that conditioning on a correct idea or bad idea polarizes the score distribution that much.
3. They provide and elaborate scaffolding for generating solutions.
4. The conducted experiments are sound. I like showing scaling of pass@k as a function of k. I like that all frontier models as well as open-weight models were used in experiments.
5. Results are complemented with an interesting analysis of the role of diversity (Fig. 5)
6. The paper is clearly written

**Weaknesses:**

1. The authors do not compare with several important baselines, e.g. iterative methods such as ReAct, Reflexion and agentic approaches (e.g. AgentCoder). I thinks that a bit weakness: there are relatively simple methods and scale well to more complex tasks (e.g. SWE-Bench).
2. I don't think it's that surprising or impressive that burning more test-time compute [1, 2, 3] leads to better results. A fair comparison would involve, e.g. a scatter plot of pass@200 on X axis and compute spent on Y axis. Compute spent can be operationalized as either tokens or dollars spent (ideally you'd report both). Then, the question is: is your method strictly optimal? Is it Pareto-optimal? See [1, 2, 3] for a discussion.

More specific points:
3. I think the opening sentence of the abstract is false as of November 2024: "While scaling training compute has led to remarkable improvements in large language models (LLMs), scaling inference compute has not yet yielded analogous gains". See again [1, 2, 3].
4. I don't understand this sentence: "LLMs as chatbots, in which models are oftentimes optimized to produce a single correct answer (Rafailov et al., 2024; Ouyang et al., 2022)." RL optimizes for reward (which could be correctness) and DPO optimizes a contrastive loss (e..g. preferring correct responses over incorrect ones). Neither optimizes for a single correct answer. This could be the case for supervised fine-tuning though.


[1] Large Language Monkeys: Scaling Inference Compute with Repeated Sampling
[2] An empirical analysis of compute-optimal inference for problem-solving with language models
[3] OpenAI O1 system card
[4] AI Agents That Matter

**Questions:**

See weaknesses

---

> ### Author Response · Authors · 2024-11-23
> **Response to Reviewer 2 (1/3)**
>
> Note: we added one appendix (Appendix V) at the end of the paper addressing agent baselines. The rest of the paper is identical. We plan to address the sentence-level changes in our next revision.
>
> We thank you for your detailed response, especially for your critical questions. We hope to address all your points below. We answer questions in a different order for clarity, flow, and better understanding of the underlying philosophy of our paper.
>
> # Test-time Compute
>
> > I don't think it's that surprising or impressive that burning more test-time compute [1, 2, 3] leads to better results. A fair comparison would involve, e.g. a scatter plot of pass@200 on X axis and compute spent on Y axis. Compute spent can be operationalized as either tokens or dollars spent (ideally you'd report both). Then, the question is: is your method strictly optimal? Is it Pareto-optimal? See [1, 2, 3] for a discussion.
>
> Thank you for the insightful comment. This is a common idea that is brought up upon reading our paper. First, we do report compute-normalized results in our paper as seen in Figure 17, which shows a large margin between PlanSearch and Repeated Sampling at medium-to-high tokens spent. While PlanSearch is indeed more expensive in terms of computation per sample generated, **it outperforms baselines even when this is taken into account**.
>
> This suggests perhaps a more interesting question which is the primary focus of the paper. If you *can only evaluate k times*, how would you optimize your chances of having a successful evaluation (with tractable compute)?
>
> This question is not simple to answer, and requires more thought beyond algorithms that optimize “given X amount of compute.” In most real-world cases, evaluation is often the bottleneck over the precise amount of compute spent. Thus, in many applications (including this paper, where *most of the compute time was actually spent running the generated codes*), the amount of LLM inference used is not the main structural bottleneck.
>
> By focusing on this question, we realize an interesting result; naive inference-time compute strategies hit diminishing returns rather quickly; if you want to push more compute into the strategy, the rate of pass@k increase (i.e., increase of pass@k per unit increase of compute) diminishes rather quickly as you scale inference-time compute.
>
> So, if you want to spend more inference-time compute (which is common in applications today given how efficient LLM inference is), feeding the extra compute into current algorithms will only give you slight performance boosts, no matter how much extra compute you have. Our method makes it such that **you can push in much more compute without plateauing performance** than you otherwise would have been able to.
>
> Therefore, if you only have enough compute to generate two or three solutions, naive sampling works well. However, existing algorithms do not scale well, and so if your inference-time compute budget reaches on the order of 5 to 10 solutions or more, these algorithms will begin to plateau while PlanSearch will still generate significant improvements, as seen in Figure 13 (although all result figures also show this).

---

> ### Author Response · Authors · 2024-11-23
> **Response to Reviewer 2 (2/3)**
>
> # Agentic Baselines
>
> > The authors do not compare with several important baselines, e.g. iterative methods such as ReAct, Reflexion and agentic approaches (e.g. AgentCoder). I thinks that a bit weakness: there are relatively simple methods and scale well to more complex tasks (e.g. SWE-Bench).
>
> Thank you for the criticism.
>
> Following your advice, we ran a new baseline ‘agentic’ model which generates code, runs the code on public tests, and then iterates. If there was an error in the public tests, the error is returned to the LLM, as well as the buggy code for fixing. We do wish to note that these agentic models operate in a different paradigm compared to our standard baselines, since the agentic models are given both 1) the exact public test cases in code and 2) liberal usage of code execution, both of which are extremely powerful (see Appendix R and all public test filtering figures), but a model may not have access to these in other tasks.
>
> We draw two comparisons here. In variant 1, the agentic model stops iterating if the code is successful or if the number of iterations exceeds $T=10$. Only the last code returned (either successful or exceeding iteration limit) is finally submitted to the judging environment and counted for pass@k. We compare variant 1 with PlanSearch with public test filtering, since in both methods, all the faulty submissions which do not pass public tests are discarded unless there are none that pass public tests.
>
> In variant 2, the agentic model continues iterating even if the code is successful. If the code is successful and uses $i$ iterations out of $T$, the agentic model starts over from scratch, except with $T-i$ total iterations instead. In this case, we submit all codes generated by the model to be counted for pass@k. We compare variant 2 with default PlanSearch, since for both methods, submissions are not guaranteed to pass public tests. This is still slightly unfair to PlanSearch baselines, since the last submissions of the agentic model will have already iterated and extracted sufficient signal from public test execution, whereas PlanSearch does not use any test execution signal at all.
>
> These plots can be found in the new Appendix V. Figure 42 corresponds to variant 1, and Figure 43 corresponds to variant 2.
>
> As seen in the plots for variant 1, the agentic model after 10 steps outperforms naive Repeated Sampling (with no public test filtering) by a margin of 0.05 as expected at pass@1. However, the pass@1 of the agentic model is not comparable to Repeated Sampling with public filtering.
>
> PlanSearch with public filtering, in both GPT-4o-mini and GPT-4o, is **strictly better than the agentic model at all k with large margins** (between $0.1$ and $0.2$). Even without public filtering, basic PlanSearch still beats variant 1 past k=3, even though basic PlanSearch has zero environment interaction.
>
> Shifting our analysis to variant 2 plots, we find similar results. As expected, without the filtering the generations of the agentic model by returning only the last generation, the agentic model roughly tracks naive Repeated Sampling, performing marginally better at large k. However, **PlanSearch still outperforms with a significant margin** past k on the order of 2 to 3.
>
> # Abstract Revision
>
> > I think the opening sentence of the abstract is false as of November 2024: "While scaling training compute has led to remarkable improvements in large language models (LLMs), scaling inference compute has not yet yielded analogous gains". See again [1, 2, 3].
>
> We appreciate your comment and attentiveness. Due to recent events, this abstract is now likely indeed out of date. We plan on editing the sentence to say: “While scaling training compute has led to remarkable improvements in large language models (LLMs), scaling inference compute *only very recently began* to yield analogous gains…”

---

> ### Author Response · Authors · 2024-11-23
> **Response to Reviewer 2 (3/3)**
>
> # RL/DPO Comment
>
> > I don't understand this sentence: "LLMs as chatbots, in which models are oftentimes optimized to produce a single correct answer (Rafailov et al., 2024; Ouyang et al., 2022)." RL optimizes for reward (which could be correctness) and DPO optimizes a contrastive loss (e..g. preferring correct responses over incorrect ones). Neither optimizes for a single correct answer. This could be the case for supervised fine-tuning though.
>
> We are happy to rephrase this section in the next revision. Our point was merely to convey that RL post-training methods do not place very much emphasis on generating a diverse set of correct answers. This is because a single correct answer with high reward would satisfy the RL constraint of maximizing reward. DPO is slightly better in this regard in that all correct responses are incentivized by the contrastive loss to rank correct answers higher than incorrect answers, but appears that it reduces diversity similarly to RL [1]. In any case, we will rephrase the above to be more clear.
>
> Again, we thank you for your detailed review. We hope to have addressed your concerns, but please do let us know if you have any remaining questions.
>
> [1] One fish, two fish, but not the whole sea: Alignment reduces language models’ conceptual diversity

---

> > ### Comment · Reviewer_CksG · 2024-11-24
> >
> > Thanks for the detailed response! I updated my score.
> >
> > > you can push in much more compute without plateauing performance
> >
> > I'd suggest making this perspective more prominent in the paper.

---

### Official Review · Reviewer_DNQT · 2024-11-02

**Soundness:** 4
**Presentation:** 4
**Contribution:** 4
**Rating:** 10
**Confidence:** 3

**Summary:**

This paper investigates the effect of inference-time search in LLMs applied to contemporary coding benchmarks. The investigations start by asking about the appropriate level of abstraction at which to perform search, and identifies "natural language plans" as a useful level through clear experimentation. They introduce a new inference-time search algorithm for LLMs in code generation called "PlanSearch". The algorithm appears general enough to be applied to other domains, though the paper does not pursue this. In the main experiments, the authors find that PlanSearch improves LLM performance on coding benchmarks significantly, and by a large margin compared to standard inference-scaling methods (RepeatedSampling). Further experiments identify "idea diversity" as the key driving factor for PlanSearch's success, with their custom measures of idea diversity being predictive of downstream performance. The authors further discover that instruction-finetuned models can have less diverse ideas than base model variants, raising important questions around the best way to perform LLM post-training.

**Strengths:**

There were a great many things I liked about this paper, and I learned a lot from its experiments.
- Figure 1 is very compelling and sells the main takeaway immediately: Searching over *ideas* in natural language (PlanSearch) is a significantly more effective way to spend inference-time compute than naive repeated sampling of direct solutions.
- The broad strategy of "generate plan first then generate the solution based on that" is highly applicable to other domains; PlanSearch can be used to generate plans for anything, and could plausibly improve performance across many domains besides code generation.
- Useful takeaway that performance can be seen as a function of diversity of ideas. This is an important lesson for the field which is not currently prioritizing LLMs' diversity of ideas, but should take idea diversity more seriously given this evidence.
- Interesting to see that instruction-tuned models can sacrifice the diversity of ideas present in base models, Figure 30 is a great figure illustrating this effect! This line was quite surprising to me:
  > in many cases, despite instruction tuned models outperforming base models by large margins on a single sample regime (pass@1), this trend disappears—sometimes even reversing—on a multi-sample regime (pass@k). We refer to Figure 30 as an example of this phenomenon

  and this further line in the Conclusion is clear-sighted in pointing out the implication of the problem of losing diversity during post-training:
  > while PLANSEARCH substantially improves diversity over idea space at inference-time, fundamentally, improvements in diversity should also come at the post-training stage
- Section 3 builds a great foundation to motivate why we care about searching over idea sketches: Starts by considering the correct layer of abstraction to search over, run experiments showing the power of the right "idea sketch". Figure 3a and 3b are great.
- Interesting to see that o1-mini, a model which itself already scales inference-time compute, benefits less from this method (which makes sense, but good to know).
- Impressive thoroughness in sharing results, with >40 figures throughout the paper and appendices!

Overall, the paper is very well-written, adds new insights to an emerging topic (diversity and search) with important ramifications for the field, and is very thorough with well-designed experiments, with many interesting results.

**Weaknesses:**

Few complaints overall. One minor weakness that doesn't cut against the broader claims and lessons of the paper:
- The actual search algorithm of PlanSearch in Section 4.3 feels somewhat arbitrary. If the goal is simply to generate diverse plans, I suspect there will be many other different ways to prompt LLMs to generate diverse ideas besides the specific PlanSearch algorithm as described. Did the authors try other algorithms / prompts? Would have been nice to see the failure cases and understand why this specific design was selected.
  - Ablations in Appendix H address this complaint somewhat, but still assumes the same algorithm structure and is mostly just a "hyperparameter search" which has small effect on the results.
  - To give a concrete example of what I imagine could be a completely different approach: directly prompting models with previously sampled ideas, and asking models to generate different plans.

**Questions:**

- > Interestingly, we find that IDEASEARCH performs somewhat better, which we speculate comes from differences in splitting solution sketch into two model responses, instead of doing both chain-of-thought and code solution in one model response.

  This is surprising. Is the only difference here that a new model response comes as a new "message" in the chat LLM API?

---

> ### Author Response · Authors · 2024-11-23
> **Response to Reviewer 1**
>
> We appreciate the detailed feedback and kind review.
>
> # PlanSearch Design
>
> Regarding the design of PlanSearch, we explore several ablations on the design of PlanSearch in Appendix H. Despite this, we agree with the reviewer that there are likely even more optimal ways of eliciting diversity and exploration besides the concrete implementation of PlanSearch.
>
> For example, when developing PlanSearch, we considered using previous ideas in sequence and prompting an LLM to produce ‘different’ generations. In fact, in Section 4.3.3, our algorithm supposes a generated idea is incorrect—adding another set of generations—which does indeed increase our pass@k by roughly 3 percent. However, as you scale up, you will need to either keep all past generations within the context length (and thus the LLM has a harder time to generate truly different ideas) or truncate at some point, which runs the risk of repeating. Therefore, we were unable to scale the technique suggested beyond the order of 5 or 10 generations. In addition, such sequential generations are slightly less efficient due to lack of batching.
>
> # IdeaSearch vs CoT
>
> Regarding the source of differences between IdeaSearch and CoT. Indeed, the main difference between IdeaSearch and CoT is that IdeaSearch splits the idea and implementation into two model responses (ignoring subtle prompt differences). We believe that having a new message is surprisingly significant. One possible explanation for this phenomenon is that IdeaSearch allows the LLM to better follow the solution sketch. In the limit case, the prompt could be structured as “XYZ idea is guaranteed to be correct…” Whereas in CoT, the model is implicitly trained to follow its own chain-of-thought, but is not as bound to the chain-of-thought as IdeaSearch. Note that this is one possible explanation; further conclusions would be speculation, and we leave investigation of this interesting phenomenon to future work.
>
> We thank you again for your feedback and insightful questions. We hope our responses clarify the points raised and look forward to any further discussions.

---

### Meta-Review · Area_Chair_VH7f · 2024-12-21

**Metareview:**

The paper introduces PLANSEARCH, a novel search algorithm that enhances code generation by leveraging natural language planning. The authors identify a lack of diversity in large language model (LLM) outputs during inference, leading to inefficient search processes. To address this, PLANSEARCH generates diverse observations about a problem and constructs natural language plans before producing code solutions. The authors reported strong empirical results across benchmarks like HumanEval+, MBPP+, and LiveCodeBench. There is some minor feedback, including the discussion of test-time compute cost and the comparison with self-improve baselines and agentic methods. The authors should try to incorporate the feedback into the final paper.

**Additional Comments On Reviewer Discussion:**

The points raised by the reviewers:
- Baseline Comparisons: Lack of comparisons with related methods (ReAct, ReFlexion, AgentCoder).
- Compute Costs: Questions about whether the observed improvements justify the additional inference compute.
- Algorithm Design: PLANSEARCH's design appears somewhat arbitrary; reviewers suggest alternative approaches could be equally valid.

During the rebuttal, the authors did well to address the above concerns with comprehensive experiments and derived valuable insights for the community. Minor refinements were suggested, including better discussion on compute cost optimization.

---

### Decision · Program_Chairs · 2025-01-22

Accept (Spotlight)